# Learning Heterogeneous Degradation Representation for Real-World Super-Resolution

**Haowei Li[1], Pengxu Wei[1,2]\*, Dongyu Zhang[1], Liang Lin[1,2]**
[1]School of Computer Science and Engineering, Sun Yat-sen University [2]Peng Cheng Laboratory
`lihw59@mail2.sysu.edu.cn`, `{weipx3,zhangdy27}@mail.sysu.edu.cn`,
`linliang@ieee.org`

## Abstract

Real-World Super-Resolution (RWSR) aims to reconstruct high-resolution images from low-resolution inputs captured under complex, real-life conditions, where diverse distortions result in significant degradation heterogeneity. Many methods rely on degradation representations, yet they struggle with the lack of spatially variant degradation modeling and degradation-content entanglement. We propose Spatially Amortized Variational Learning (SAVL), an implicit framework that models per-pixel degradations as spatially varying Gaussians inferred from local neighborhoods. SAVL couples a conditional likelihood lane (SAVL-LM) with a mutual information suppression lane (SAVL-MIS) to filter out degradation-irrelevant signals, yielding a well-constrained solution space. Both our qualitative visualizations and quantitative analyses confirm that the learned representations effectively capture the spatial distribution of complex degradations while being highly discriminative of diverse underlying degradation factors. Building on these representations, we design a degradation-aware SR network with channel-wise guidance and spatial attention modulation for adaptive reconstruction under heterogeneous degradations. Extensive experiments on real-world datasets demonstrate consistent gains over prior methods.

## 1 Introduction

Single Image Super-Resolution (SISR) aims to reconstruct a high-resolution image from a single low-resolution input by recovering lost high-frequency details and structural information. Traditional SISR methods often rely on a fixed and idealized degradation model, most commonly bicubic downsampling Dong et al. (2015); Ledig et al. (2017); Lim et al. (2017); Wang et al. (2018b); Zhang et al. (2018c). However, this assumption fails to capture the heterogeneous degradations present in real-world scenarios, leading to significant performance drops. Real-world image degradations are substantially heterogeneous, varying both between images and spatially within a single image. This heterogeneity stems from a wide array of factors, including differences in imaging devices Ignatov et al. (2017); Wei et al. (2020); Xu et al. (2022), varying capture conditions Aakerberg et al. (2024) (e.g., ISO), and diverse ISP pipelines, as well as intrinsic image properties like DoF and texture complexity Kim et al. (2021); Liang et al. (2021b); Chen et al. (2023b) that introduce spatially non-uniform degradations.

To tackle complex real-world degradations, many works learn a degradation representation to guide the image upsampling process. These approaches can be broadly categorized into two main paradigms: explicit degradation estimation and implicit degradation learning. *Explicit degradation estimation* Gu et al. (2019); Liang et al. (2021b); Huang et al. (2020); Liang et al. (2022) operates by defining a parametric degradation space. These methods typically synthesize data with a fixed degradation pipeline and then supervise an estimator to predict specific degradation parameters (e.g., blur kernel, noise level). However, the reliance on a predefined and often simplistic degradation space limits their applicability to real-world scenarios, where degradations are more complex and deviate from the training assumptions. In contrast, *implicit degradation learning* aims to construct a latent

---

*Corresponding author.

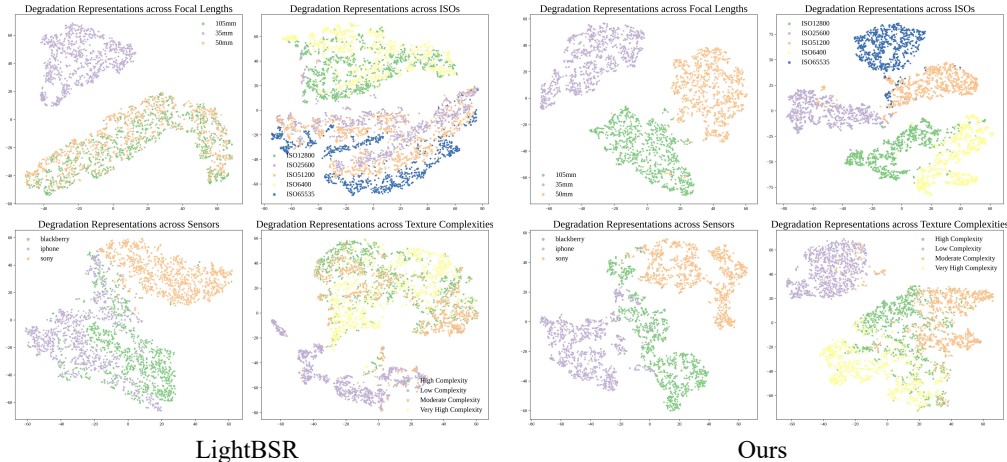

LightBSR                                    Ours

Figure 1: Visualization of Degradation Representations. Compared to LightBSR Yuan et al. (2025), our learned representations form well-separated clusters for distinct degradation factors (ISO, focal length, sensor, and texture). This demonstrates a superior ability to discriminate between fine-grained degradation variations caused by these factors.

space for an Implicit Degradation Representation (IDR) that can generalize to unknown and complex degradations. Rather than predicting fixed parameters, these methods employ techniques such as meta-learning Xia et al. (2023), contrastive learning Wang et al. (2021a); Mou et al. (2022); Lin et al. (2024a), or knowledge distillation Xia et al. (2022); Liu et al. (2024); Yuan et al. (2025) to learn deterministic IDRs that encode diverse degradation patterns.

Existing Implicit Degradation Representations (IDRs) have two major shortcomings: (1) *Lack of modeling for spatially variant degradations.* They often fail to account for the fact that real-world complex degradations are not uniformly distributed within an image. (2) *Insufficient decoupling of degradation from content.* Unlike the constrained parametric spaces of explicit estimation, the latent space for IDRs is complex and unconstrained. This high capacity allows weakly regularized embeddings to be overly expressive, easily encoding the entire LR signal, including degradation-irrelevant content (e.g., appearance, semantics) Li et al. (2024); Yuan et al. (2024). This degradation-content entanglement renders the representation non-discriminative for degradation, failing to provide useful guidance for the SR process and thus impairing generalization. For instance, as shown in Fig. 1, representations from LightBSR Yuan et al. (2025) show significant overlap across different degradation factors, whereas our learned representations form distinct clusters corresponding to factors such as focal length, ISO, sensor, and texture.

The challenge of degradation-content entanglement Wei et al. (2023b); Guo et al. (2024b); Wei et al. (2023a) is magnified when modeling spatially variant degradations. A per-pixel latent space is inherently more complex and, without effective regularization, becomes more prone to encoding content. Moreover, the assumption of spatial variance directly undermines contrastive learning meth-

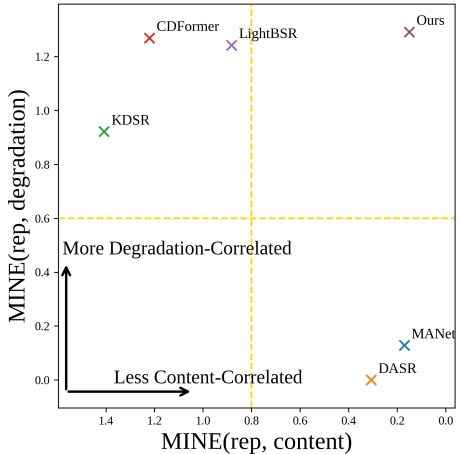

Figure 2: Correlation of IDR with Degradation and Content. IDRs are mapped by correlation with degradation factors (Y-axis) and image content (X-axis): explicit representations (bottom-right) show low correlation with both, while conventional implicit ones (top-left) are highly correlated with both, revealing strong degradation–content entanglement.

ods Yuan et al. (2024), as their core assumption that different patches from the same image share the same degradation and thus form positive pairs no longer holds.

To address these challenges, we introduce **SAVL**, a Spatially Amortized Variational Learning framework. SAVL models per-pixel degradations as spatially varying Gaussian distributions inferred from local image neighborhoods. The Gaussian posteriors inherently capture the spatial non-uniformity of degradation. To decouple the representation from content, we enforce a whitening prior and apply an explicit mutual information constraint. We leverage an amortized inference network to efficiently predict the local posteriors, thereby avoiding the prohibitive computational cost of per-pixel optimization Guo et al. (2024a). For each pixel, the predicted posterior provides a mode and a variance; the mode characterizes the degradation type, while the variance quantifies its uncertainty or severity, yielding an interpretable representation. This dual-component representation enables a powerful guidance mechanism: the mode provides channel-wise modulation, and the variance provides spatial modulation for the downstream SR network. Our approach achieves state-of-the-art results on both synthetic and real-world datasets. Fig. 2 visualizes the correlation of various degradation representations with both degradation factors and image content, quantified via Mutual Information Neural Estimation (MINE) Belghazi et al. (2018) (details in Sec. 4.2). The analysis reveals a clear trade-off. Explicit representations, found in the bottom-right quadrant, successfully suppress content correlation due to their constrained parametric nature, but at the cost of insufficient degradation modeling capability. Conversely, existing implicit representations populate the top-left quadrant, exhibiting a strong correlation with both axes and thus indicating severe degradation-content entanglement. In contrast, our method is the only approach that learns an implicit representation with strong degradation correlation while simultaneously minimizing content correlation. In summary, our main contributions are as follows:

- **Spatial heterogeneity modeling.** We introduce SAVL, a framework that learns a spatially varying Gaussian posterior for each pixel from its local neighborhood. This method achieves stable, spatially-resolved degradation representation in a single, efficient pass.
- **Suppressing degradation-content entanglement.** We incorporate a mutual-information suppression term into the conditional ELBO, which explicitly filters degradation-irrelevant content. The result is a per-pixel representation that is both well-constrained and highly discriminative of the underlying degradation.
- **Degradation-aware SR integration.** We propose a novel mechanism that injects our representation into an SR backbone using both components of the learned posterior: the mode provides channel-wise guidance, and the variance enables spatial feature modulation. This strategy delivers consistent performance gains across synthetic and real-world benchmarks.

## 2 Related Work

**Real-World Super-Resolution.** Traditional SR methods often assume a simplistic degradation like bicubic downsampling Dong et al. (2015); Ledig et al. (2017), hindering their performance on real-world images. While datasets with realistic paired images like RealSR Cai et al. (2019) and DRealSR Wei et al. (2020) have been developed to mitigate this, a major research direction involves explicitly modeling the complex degradations. In the non-blind setting, methods like SRMD Zhang et al. (2018a), UDVD Xu et al. (2020), and DPSR Zhang et al. (2019) take known degradation parameters as input. For the more challenging blind setting, numerous approaches estimate these parameters first. These include supervised estimation of global degradations Gu et al. (2019); Liang et al. (2021b; 2022), estimation of spatially varying degradation Kim et al. (2021); Chen et al. (2023b); Guo et al. (2024a), and joint optimization of the estimator and the SR network Huang et al. (2020); Cai et al. (2019); Guo et al. (2024a). Despite their progress, all of these explicit estimation strategies are confined to a predefined degradation space. This limits their generalization, as real-world degradations are often more complex and unknown at training time, posing a significant challenge to existing solutions.

**Implicit Degradation Representation Learning.** Numerous methods utilize implicit degradation representation learning to guide the SR process. Foundational approaches established various learning paradigms to obtain supervised or self-supervised signals, including contrastive learning Wang et al. (2021a), knowledge distillation Xia et al. (2022); Yuan et al. (2025), and metric learning Mou et al. (2022). Subsequent works have focused on refining these representations, for instance, by

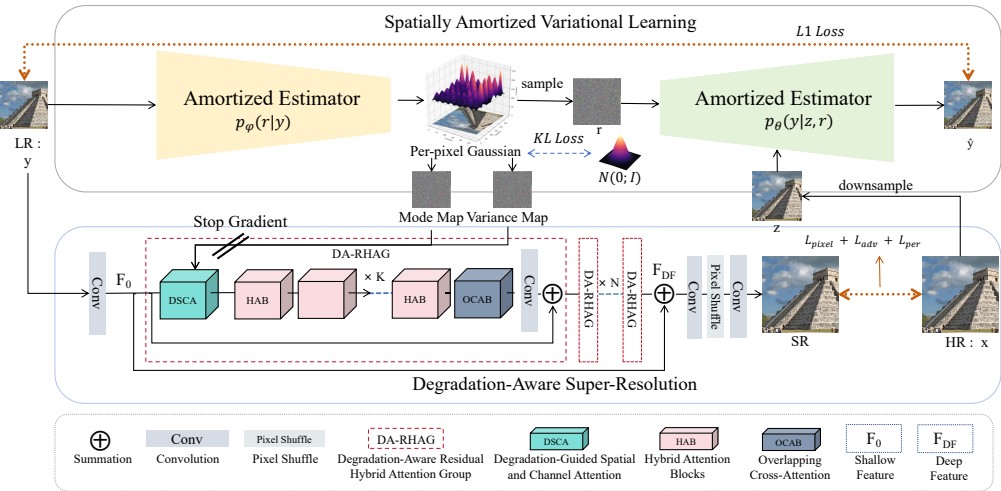

Figure 3: The main pipeline of our method.

modeling them as probability distributions to suppress uncertainty Lin et al. (2024a) or by leveraging powerful generative ability like diffusion models Liu et al. (2024). A central challenge in this domain is the entanglement of degradation and content features. To address this, some methods propose explicit decoupling mechanisms, such as cyclic shift sampling Yuan et al. (2024). Another challenge is to learn spatially variant degradation from real data. Aakerberg et al. (2024) learns a pixel-wise degradation feature, though simple additive formulations without strong priors risk content leakage into the representation. Despite this progress, a fundamental tension remains: the push towards modeling spatially variant degradations requires a more complex, per-pixel latent space, which in turn makes the goal of robustly decoupling degradation from content features significantly more challenging.

## 3 SPATIALLY AMORTIZED VARIATIONAL LEARNING

**Overview.** We present **SAVL** (*Spatially Amortized Variational Learning*) to learn heterogeneous degradations. SAVL learns degradations *implicitly*: each pixel-wise degradation variable is modeled as a *spatially varying Gaussian* whose posterior is inferred from local evidence with parameters shared across space and images. The framework has two lanes that are estimated by the same amortized networks and composed at the end. *SAVL-LM* learns a spatially factorized conditional likelihood via a conditional ELBO. *SAVL-MIS* implements a mutual-information *suppression* strategy that upper-bounds and penalizes $I(r; z)$, ensuring that the learned degradation distribution captures degradation-specific factors rather than content. Sharing estimators across lanes yields a two-term loss and single-pass inference. As shown in Fig. 3, SAVL and the SR network are jointly trained at first. The learned representations are directly injected into the SR network for degradation-aware super-resolution. We insert our Degradation-Guided Spatial and Channel Attention (DSCA) module illustrated in Fig. 4 before Hybrid Attention Blocks (HAB) and Overlapping Cross-Attention (OCAB) Chen et al. (2023a), serving as the initial stage of every Degradation-Aware Residual Hybrid Attention Group (DA-RHAG). Once converged, Amortized Estimators are frozen for downstream SR tasks. More details are provided in Section 3.4.

### 3.1 SETUP AND ASSUMPTIONS

Let $x$ be a clean high-resolution image and $y$ the observed low-resolution image. Let $D$ denote a deterministic downsampling operator (e.g., bicubic), and define the content code $z = D(x)$. Let $\Omega \subset \mathbb{Z}^2$ index pixels, and let $\mathcal{N}_s(u)$ be the Chebyshev neighborhood of pixel $u$ with radius $s$. Let $r(\cdot) = \{r(u) \in \mathbb{R}^m\}_{u \in \Omega}$ denote a spatially varying degradation field.

**Learning objective from paired data.** We learn degradations from paired real data $(x, y)$, so we can optimize the conditional marginal likelihood of $y$ given $z$ under the data distribution $p_{\text{data}}(x, y)$. To prevent content leakage into the degradation variables, we constrain the mutual information between $r$ and $z$:

$$\max \ \mathbb{E}_{p_{\text{data}}(x,y)}\big[\log p_\Theta(y \mid z)\big] \quad \text{s.t.} \quad I(r; z) \leq \kappa. \tag{1}$$

$I(r; z)$ is taken under $(r, z) \sim p_{\text{data}}(x, y)\, q(r \mid y)$ with $z = D(x)$. We collect all parameters as $\Theta$. Using a Lagrange multiplier $\lambda \geq 0$, we obtain the penalized objective

$$\mathcal{L}(\Theta, \lambda) = \mathbb{E}_{p_{\text{data}}(x,y)}\big[\log p_\Theta(y \mid z)\big] - \lambda\, I(r; z), \tag{2}$$

where the additive constant $\lambda\kappa$ is dropped since it does not depend on $\Theta$. We address the two terms in this penalized objective separately. **SAVL-LM** maximizes a tractable lower bound on the log-likelihood term, while **SAVL-MIS** minimizes a tractable upper bound on the mutual information term.

**Assumptions.** **(A1) Spatial amortization.** Both the likelihood $p_\theta(y \mid z, r)$ and the variational posterior $q_\psi(r \mid y)$ are implemented using receptive-field–limited networks Liang et al. (2021b) with parameters shared across pixels and images, yielding amortized inference that replaces costly per-image and per-pixel optimization Guo et al. (2024a) with a learned single-pass mapping. This design suits real-world degradations: optical effects, sensor noise, and compression act locally and vary smoothly, so shared local modules capture a common rule across space and images, reducing variance and parameter count. **(A2) Local conditional independence.** Given $(z, r)$, pixels are conditionally independent and each pixel depends only on local content and local degradation:

$$p_\theta(y \mid z, r) = \prod_{u \in \Omega} p_\theta\big(y(u) \mid z(\mathcal{N}_s(u)),\, r(u)\big). \tag{3}$$

**(A3) Mean-field Gaussian posterior and white prior.** We adopt a mean-field Gaussian posterior with a spatially white Gaussian prior:

$$q_\psi(r \mid y) = \prod_{u \in \Omega} \mathcal{N}\big(r(u);\, \mu_\psi(u),\, \text{diag}\, \sigma_\psi^2(u)\big), \qquad p(r) = \prod_{u \in \Omega} \mathcal{N}\big(r(u);\, \mathbf{0}, \mathbf{I}\big), \tag{4}$$

$$\mu_\psi(u),\, \log \sigma_\psi^2(u) = g_\psi\big(y(\mathcal{N}_s(u))\big), \qquad r(u) = \mu_\psi(u) + \sigma_\psi(u) \odot \varepsilon(u), \quad \varepsilon(u) \sim \mathcal{N}(\mathbf{0}, \mathbf{I}). \tag{5}$$

This yields an implicit, spatially varying Gaussian family for degradations, inferred from local evidence via shared, receptive-field–limited networks.

## 3.2 SAVL-LM: Conditional ELBO

To address the log-likelihood term $\mathbb{E}[\log p_\Theta(y \mid z)]$ in equation 2, we maximize its evidence lower bound (ELBO). For any variational posterior $q_\psi(r \mid y)$, Jensen's inequality yields

$$\log p_\Theta(y \mid z) \geq \mathcal{L}_{\text{LM}}(\theta, \psi) := \mathbb{E}_{q_\psi}\big[\log p_\theta(y \mid z, r)\big] - D_{\text{KL}}\big(q_\psi(r \mid y) \,\|\, p(r)\big). \tag{6}$$

A derivation of equation 6 is provided in Appendix A.1. Under equation 4, the Kullback–Leibler term has the closed form

$$D_{\text{KL}}\big(q_\psi \| p\big) = \tfrac{1}{2} \sum_{u \in \Omega} \sum_{j=1}^{m} \Big(\mu_{\psi,j}(u)^2 + \sigma_{\psi,j}^2(u) - \log \sigma_{\psi,j}^2(u) - 1\Big), \tag{7}$$

## 3.3 SAVL-MIS: Mutual-Information Suppression

To address the mutual information term $I(r; z)$ in equation 2, our goal is to derive and minimize a tractable upper bound, thereby suppressing the dependence of the degradation variables $r$ on the content code $z$. Starting from the identity

$$I(r; z) = I(r; y) + I(z; y) - I(y; z, r) + C, \tag{8}$$

where $C$ is a constant, we note that $I(z; y)$ does not involve any learnable parameters and is therefore omitted in optimization. A detailed derivation of equation 8 is provided in Appendix A.2. We control $I(r; z)$ by (i) an upper bound on $\text{I}(r; y)$ using a VIB-style term Alemi et al. (2019) and (ii) a lower bound on $\text{I}(y; z, r)$ via Barber–Agakov term Barber & Agakov (2004).

Figure 4: DSCA: Degradation-Guided Spatial and Channel Attention.

**Upper and lower bounds.** For any $p'(r)$ and $q_\phi(r \mid y)$,

$$I(r; y) \leq \mathbb{E}_y\Big[D_{\mathrm{KL}}\big(q_\phi(r \mid y) \,\|\, p'(r)\big)\Big] + C_1, \tag{9}$$

and for any critic $p_\vartheta(y \mid z, r)$,

$$I(y; z, r) \geq \mathbb{E}_{q_\phi(r|y)}\big[\log p_\vartheta(y \mid z, r)\big] + C_2, \tag{10}$$

where $C_1$ and $C_2$ are constants independent of the learnable parameters. Combining equation 8–equation 10 yields the MIS penalty (up to an additive constant $C'$):

$$I(r; z) \leq \mathbb{E}_y\Big[D_{\mathrm{KL}}\big(q_\phi(r \mid y) \,\|\, p'(r)\big)\Big] - \mathbb{E}_{q_\phi(r|y)}\big[\log p_\vartheta(y \mid z, r)\big] + C'. \tag{11}$$

**Alignment with SAVL-LM.** To obtain an estimator-sharing form, we finally *unify the three components*: (i) reuse the likelihood as the critic by setting $\vartheta \equiv \theta$, (ii) share the amortized posterior by setting $\phi \equiv \psi$, and (iii) adopt the same spatially white prior as in equation 4 by setting $p'(r) \equiv p(r)$. With these identifications, constants are dropped and the composed objective reduces to

$$\mathcal{J}_{\mathrm{SAVL}}(\theta, \psi) = -\mathbb{E}_{q_\psi}\big[\log p_\theta(y \mid z, r)\big] + (1 + \lambda)\, D_{\mathrm{KL}}\big(q_\psi(r \mid y) \,\|\, p(r)\big), \tag{12}$$

where the reconstruction and KL terms are estimated by the same amortized networks and neighborhoods as in SAVL-LM. In practice, we model the conditional likelihood per pixel as either Gaussian or Laplace, so that the reconstruction term reduces to an $L_2$ or $L_1$ loss, respectively:

$$\mathcal{L}_{\mathrm{rec}} = \begin{cases} \sum_{u \in \Omega} \|y(u) - \hat{y}_\theta(u; z, r)\|_2^2, & \text{(Gaussian)} \\ \sum_{u \in \Omega} \|y(u) - \hat{y}_\theta(u; z, r)\|_1, & \text{(Laplace)}. \end{cases} \tag{13}$$

In practice we use tunable weights:

$$\min_{\theta, \psi} \mathcal{L}_{\mathrm{train}} = \alpha\, \mathcal{L}_{\mathrm{rec}}(y; \theta, \psi) + \beta\, D_{\mathrm{KL}}\big(q_\psi(r \mid y) \,\|\, p(r)\big), \tag{14}$$

with $\alpha > 0$ and $\beta \geq 0$.

### 3.4 LEVERAGING LEARNED REPRESENTATIONS IN SUPER-RESOLUTION

Building upon the utility of attention in super-resolution Liang et al. (2021a); Zhou et al. (2023); Chen et al. (2023a), we integrate our degradation representation, $r$, via a Degradation-Guided Spatial–Channel Attention (DSCA) module (Fig. 4). DSCA leverages the two components of the per-pixel Gaussian posterior $r$ for a dual-modulation strategy: the posterior variance guides spatial attention, while the posterior mode (mean) guides channel attention.

This is implemented as follows. For spatial attention, we define degradation severity as the posterior variance to reweight SW-MSA Liang et al. (2021a) scores, promoting attention between pixels with similar degradation. For channel attention, a lightweight convolutional network uses the posterior mode to predict a channel-wise modulation vector, adjusting feature activations based on the inferred degradation type.

## 4 EXPERIMENTS

### 4.1 EXPERIMENT SETTINGS

To validate the two primary properties of our learned degradation representation, we conduct a series of analyses. First, to verify that our representation effectively captures **spatial heterogeneity**,

| Method | HSIC (Rep ↔ SceneID) ↓ | HSIC (Rep ↔ Noise-Level) ↑ | MINE (Rep, SceneID) ↓ | MINE (Rep, Noise-Level) ↑ | Scene-ID Acc. ↓ | Noise-Level Acc. ↑ |
|---|---|---|---|---|---|---|
| MANe | **0.0056** | 0.0043 | 0.1707 | 0.1286 | **0.3253** | 0.3150 |
| DASR | 0.0187 | <0.0001 | 0.3079 | <0.0001 | 0.6065 | 0.2000 |
| KDSR | 0.0134 | 0.0058 | 1.4076 | 0.9217 | 0.9346 | 0.5272 |
| CDFormer | 0.0131 | 0.0270 | 1.2209 | 1.2694 | 0.8456 | 0.8393 |
| LightBSR | 0.0059 | 0.0133 | 0.8823 | 1.2425 | 0.8392 | 0.9110 |
| **Ours** | 0.0124 | **0.0304** | **0.1507** | **1.2920** | 0.4997 | **0.9480** |
| *Ours w/o SAVL* | 0.0181 | 0.0220 | 1.0254 | 1.1240 | 0.9404 | 0.8148 |
| *Ours w/o SAVL + CLUB* | 0.0090 | <0.0001 | 0.2700 | <0.0001 | 0.3644 | 0.2000 |

Table 1: Dependency metrics and linear-probe accuracies on SVSR Aakerberg et al. (2024) with ablations. Lower association/accuracy w.r.t. Scene ID and higher association/accuracy w.r.t. noise-level indicate a better degradation representation.

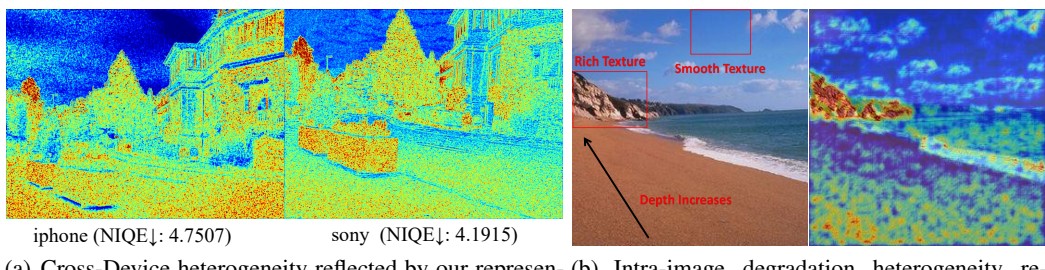

iphone (NIQE↓: 4.7507)    sony (NIQE↓: 4.1915)

(a) Cross-Device heterogeneity reflected by our representation.

(b) Intra-image degradation heterogeneity reflected by our representation.

Figure 5: Visualization of local degradation severity.

we perform a qualitative analysis using the degradation severity map introduced in Sec. 4.2. This map allows for visual inspection of the model's ability to identify spatially varying degradation patterns arising from complex factors. Second, to confirm that the representation is **decoupled from content**, we qualitatively assess the structure of the latent space via visualization. Quantitatively, we assess the representation's correlation with image content and degradation factors via linear probing Alain & Bengio (2016); Chen et al. (2020), and directly test for statistical independence using both the Hilbert-Schmidt Independence Criterion (HSIC) Gretton et al. (2007) and Mutual Information Neural Estimation (MINE) Belghazi et al. (2018).

Another aspect that needs to be validated is the performance of our degradation-aware super-resolution. We train it on the synthetic dataset from Real-ESRGAN Wang et al. (2021b), which is constructed using ground-truth images from DF2K Agustsson & Timofte (2017); Lim et al. (2017) and OutdoorSceneTraining Wang et al. (2018a). We evaluate the trained model on both synthetic and real-world benchmarks Wei et al. (2020); Cai et al. (2019); Aakerberg et al. (2024). During testing, all images are processed as $256 \times 256$ tiles, with further details provided in Sec. 4.3.

We adopt a two-stage schedule in a single pipeline: we first jointly train SAVL and the SR backbone for 200K iterations to convergence, after which we fine-tune the SR model with an $\ell_1$ loss for 800K iterations followed by $\ell_1$+perceptual+adversarial losses for an additional 600K iterations; unless otherwise noted, we use Adam ($\beta_1$=0.9, $\beta_2$=0.99), a batch size of 32, and 64×64 patches, with an initial learning rate of $2\times10^{-4}$ in $\ell_1$ phases and $1\times10^{-4}$ in adversarial phases, halved every 200K iterations; all experiments are implemented in PyTorch and executed on an $8\times$ NVIDIA RTX 3090 GPU setup. The following standard metrics are used for performance comparison: **PSNR**, **SSIM**, and **LPIPS** Zhang et al. (2018b). **More implementation details are provided in the Appendix A.4.**

## 4.2 EFFECTIVENESS OF DEGRADATION REPRESENTATION

**Visualization of Local Degradation Severity.** We define the per-pixel degradation severity by a single normalized transform of the posterior variance map:

$$s(u) = 1 - \frac{\sigma_\psi^2(u) - \mu_{\sigma_\psi^2}}{\mathrm{Var}[\sigma_\psi^2]}, \tag{15}$$

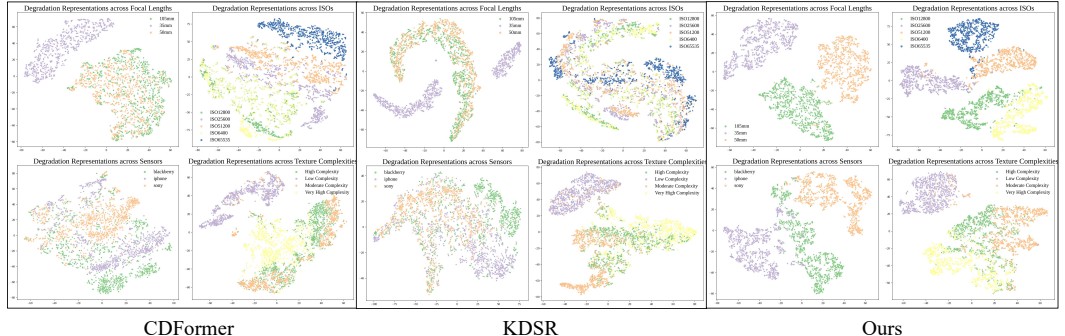

Figure 6: Visualization of the learned representation space. Our representations form well-separated clusters for distinct degradation factors, revealing strong discrimination of fine-grained variations, whereas other methods tend to collapse these factors into overlapping embeddings. In each grid, the four corners (top-left to bottom-right) correspond to changes in focal length, ISO, sensor type, and texture complexity, respectively. The results of LightBSR under the same setting are summarized in Fig. 1.

where $\sigma^2_\psi(u)$ denotes the per-pixel variance from the mean-field Gaussian posterior in equation 5, and $\mu_{\sigma^2_\psi}$ and $\mathrm{Var}[\sigma^2_\psi]$ are the image-wise or pair-wise mean and variance of $\sigma^2_\psi$, respectively.

As shown in Fig. 5, the ability to capture local degradation patterns demonstrates the spatial sensitivity and interpretability of our learned degradation representation. For cross-device heterogeneity in Fig. 5(a), using same-scene images from different devices in DPED Ignatov et al. (2017), the representation reliably reflects degradation severity: images from the iPhone, which exhibit stronger degradations, yield heatmaps that are more intense and spatially concentrated, consistent with NIQE Mittal et al. (2012) scores. For intra-image heterogeneity in Fig. 5(b), the representation reveals spatial variations in degradation severity as a function of scene depth and texture complexity.

**Assessment Across Degradation Factors.** We qualitatively assess discriminability by visualizing the learned latent space (PCA to 32D, then t-SNE to 2D) for our method versus baselines (Fig. 6).

To isolate specific degradation factors, we leverage datasets that provide captures of identical scenes while varying a single degradation source: SVSR Aakerberg et al. (2024) for ISO, RealSR Cai et al. (2019) for focal length, and DPED Ignatov et al. (2017) for camera sensors. In each of these fixed-scene settings, as well as for intra-image variations in local texture, the results show a clear distinction between our method and the baseline. While baseline representations fail to distinguish the induced variations and remain entangled, our representation exhibits a clear, degradation-driven structure, forming distinct clusters that align with the specific factor (e.g., noise level, lens setting, or complexity bin).

We then quantify these trends via linear probing and dependence analyses using the Hilbert–Schmidt Independence Criterion (HSIC) and Mutual Information Neural Estimation (MINE). We instantiate the degradation factor as ISO-induced noise level and define image content by scene identity (Scene ID). As detailed in Table 1, the results reveal a sharp **contrast in predictability**: while baselines like KDSR and CDFormer retain high accuracy for both factors ($> 83\%$ for Scene ID), SAVL maintains high degradation accuracy (**94.80%**) while suppressing content accuracy to a chance-level (**49.97%**). This effective decoupling is further validated by dependence metrics, where SAVL achieves an order-of-magnitude lower MINE score with content (e.g., **0.1507** vs. 1.2209 for CD-Former), confirming that our method achieves disentanglement through active information suppression.

### 4.3 EVALUATION ON REAL-WORLD SUPER-RESOLUTION

We train our model on the synthetic dataset provided by Real-ESRGAN Wang et al. (2021b) and evaluate on both synthetic and real-world benchmarks, including RealSR Cai et al. (2019), DRealSR Wei et al. (2020), SVSR Aakerberg et al. (2024), and RealSRSet25 Zhang et al. (2021). As shown in Table 2, our method consistently outperforms state-of-the-art approaches, **demonstrat-**

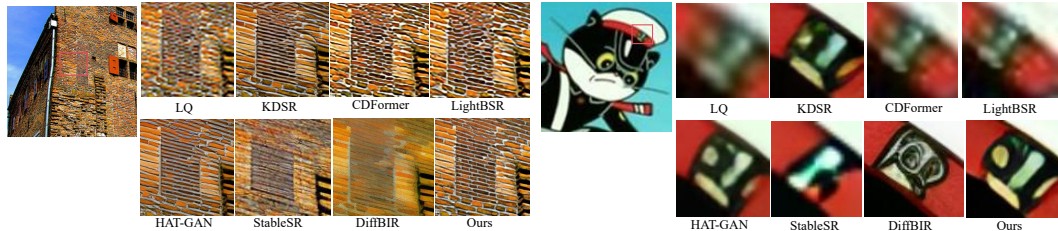

Figure 7: Qualitative comparison of SR results on the RealSet25 dataset. Our method yields more faithful texture and structural restoration while avoiding hallucinations and artifacts.

| Methods | Params (M) | GFLOPs | DIV2K | | | RealSR | | | DRealSR | | | SVSR | | |
|---|---|---|---|---|---|---|---|---|---|---|---|---|---|---|
| | | | PSNR↑ | SSIM↑ | LPIPS↓ | PSNR↑ | SSIM↑ | LPIPS↓ | PSNR↑ | SSIM↑ | LPIPS↓ | PSNR↑ | SSIM↑ | LPIPS↓ |
| BSRGAN Zhang et al. (2021) | 16.7 | 661 | 19.88 | 0.5137 | 0.4303 | 25.01 | 0.7422 | 0.2853 | 27.09 | 0.7759 | 0.2950 | 25.09 | 0.7742 | 0.3321 |
| RealESRGAN Wang et al. (2021b) | 16.7 | 661 | 19.92 | 0.5334 | 0.3981 | 24.22 | 0.7401 | 0.2901 | 26.95 | 0.7812 | 0.2876 | 24.36 | 0.7582 | 0.3584 |
| SwinIR-GAN Liang et al. (2021a) | 11.8 | 495 | 19.66 | 0.5253 | 0.3992 | 24.89 | 0.7543 | 0.2680 | 27.00 | 0.7815 | 0.2789 | 24.07 | 0.7426 | 0.3904 |
| HAT-GAN Chen et al. (2023a) | 20.8 | 960 | 19.78 | 0.5319 | 0.3917 | 25.17 | 0.7547 | 0.2611 | 27.76 | 0.7926 | 0.2744 | 25.05 | 0.7721 | **0.3007** |
| DASR Liang et al. (2022) | 8.07 | 187 | 19.73 | 0.5122 | 0.4350 | 25.51 | 0.7526 | 0.3201 | 28.19 | 0.8051 | 0.3165 | 24.53 | 0.7592 | 0.3917 |
| DiffBIR Lin et al. (2024b) | 1670 | \ | 19.98 | 0.4987 | 0.3866 | 24.94 | 0.6902 | 0.3436 | 27.16 | 0.7140 | 0.3920 | 23.97 | 0.7019 | 0.4133 |
| ResShift Yue et al. (2023) | 118.6 | \ | 19.80 | 0.4985 | 0.4450 | 24.77 | 0.7178 | 0.3864 | 27.31 | 0.7388 | 0.4101 | 24.21 | 0.7204 | 0.3980 |
| StableSR Wang et al. (2024) | 919 | \ | 19.73 | 0.5039 | 0.4145 | 24.60 | 0.7387 | 0.2736 | 27.39 | 0.7830 | 0.2710 | 24.49 | 0.7528 | 0.3815 |
| KDSR Xia et al. (2022) | 18.8 | 482 | 19.87 | 0.5361 | 0.3784 | 25.57 | 0.7588 | 0.2607 | 27.02 | 0.7787 | 0.3461 | 25.09 | 0.7891 | 0.3436 |
| CDFormer Liu et al. (2024) | 25.0 | 725 | 19.91 | 0.5302 | 0.3839 | 25.43 | 0.7550 | 0.2641 | 27.11 | 0.7792 | 0.3102 | 25.07 | 0.7862 | 0.3124 |
| LightBSR Yuan et al. (2025) | 3.1 | 206 | 19.41 | 0.5175 | 0.4650 | 24.98 | 0.7062 | 0.3445 | 27.69 | 0.7893 | 0.3004 | 24.93 | 0.7718 | 0.3308 |
| Ours | 14.0 | 499 | **20.03** | **0.5420** | 0.3778 | **25.80** | **0.7603** | **0.2455** | 28.27 | **0.8139** | **0.2650** | **25.13** | **0.7899** | 0.3181 |
| *Ours w/o Channel Modulation* | 14.0 | 499 | 19.89 | 0.5298 | 0.3901 | 25.13 | 0.7539 | 0.2832 | 27.78 | 0.7924 | 0.3010 | 24.70 | 0.7618 | 0.4026 |
| *Ours w/o Spatial Modulation* | 13.6 | 462 | 20.01 | 0.5416 | **0.3776** | 25.69 | 0.7592 | 0.2587 | 28.24 | 0.8132 | 0.2715 | 25.09 | 0.7866 | 0.3360 |

Table 2: Evaluation on synthetic and real SR datasets (×4).

**ing increasingly significant gains as data complexity rises**. While showing steady improvement on synthetic DIV2K (**+0.11 dB** over RealESRGAN), SAVL achieves a more pronounced advantage on heterogeneous real-world datasets: outperforming KDSR by **0.23 dB** on RealSR and surpassing LightBSR by **0.58 dB** on the highly complex DRealSR. Furthermore, compared to diffusion-based methods Lin et al. (2024b); Wang et al. (2024), SAVL achieves superior fidelity (e.g., SSIM **0.8139** vs. 0.7830 for StableSR on DRealSR). Qualitative results in Fig. 7 further underscore this superiority, confirming that SAVL faithfully restores texture and structural details while effectively avoiding the hallucinations and semantic artifacts often introduced by generative priors.

### 4.4 ABLATION STUDY

We ablate **SAVL** with two variants: (i) a deterministic per-pixel degradation code without any mutual-information (MI) constraint, and (ii) the same deterministic code regularized by the CLUB MI upper bound Cheng et al. (2020). On SVSR Aakerberg et al. (2024), under the protocol of Sec. 4.2, we report quantitative metrics in Table 1 and qualitative t-SNE visualizations in Fig. 8. Under fixed-scene comparisons, either ablation collapses separability across ISO (noise-level) strata. We further ablate the degradation-aware SR by removing *channel* or *spatial* feature modulations; results in Table 2 are obtained under identical training and testing protocols to isolate each component's contribution.

## 5 CONCLUSION

We introduced **SAVL**, a spatially amortized variational framework that learns a per-pixel, spatially varying degradation representation. By enforcing an information-theoretic constraint, SAVL disentangles this representation from image content, making it highly discriminative of diverse degradation factors. We leverage this representation through a novel dual-guidance mechanism that uses the posterior's mode for channel modulation and its variance for spatial modulation; this seamless integration into SR networks yields consistent performance gains on both synthetic and real-world datasets. Extensive qualitative and quantitative analyses, corroborated by ablation studies, confirm that our representation captures spatial heterogeneity while remaining strongly correlated with

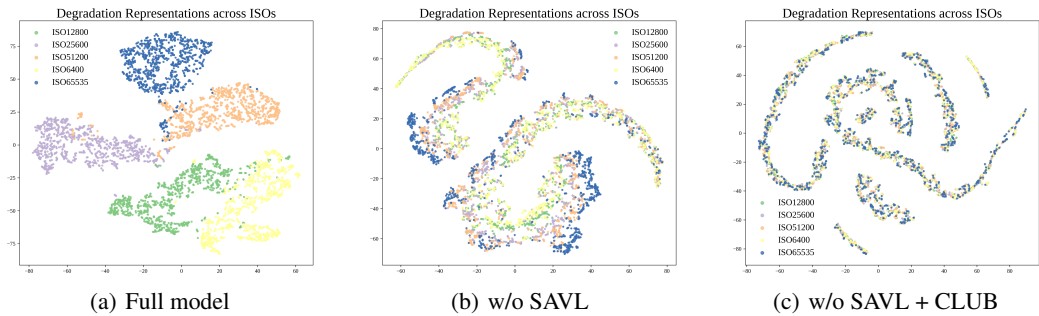

(a) Full model         (b) w/o SAVL         (c) w/o SAVL + CLUB

Figure 8: Qualitative effects of ablations (PCA+t-SNE).
.

degradation and minimally with content, and verify the essential contributions of our proposed components.

### ACKNOWLEDGMENTS

This work is supported in part by National Natural Science Foundation of China (NSFC) under Grant No.62376292, Guangdong Provincial General Fund No. 2024A1515010208, Guangzhou Science and Technology Program Project No.2025A04J5465, Guangdong Basic and Applied Basic Research Foundation Under Grant No. 2024A1515011741.

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

## A  APPENDIX

### A.1  DERIVATION OF THE CONDITIONAL ELBO

$$\log p_\theta(y \mid z) = \log \int p_\theta(y, r \mid z)\, dr \quad \text{(def.)} \tag{16}$$

$$= \log \int p(r)\, p_\theta(y \mid z, r)\, dr \quad \text{(factorize } p_\theta(y, r \mid z) = p(r)\, p_\theta(y \mid z, r)) \tag{17}$$

$$= \log \int q_\psi(r \mid y)\, \frac{p_\theta(y \mid z, r)\, p(r)}{q_\psi(r \mid y)}\, dr \quad (\times \tfrac{q_\psi}{q_\psi}) \tag{18}$$

$$= \log \mathbb{E}_{q_\psi(r\mid y)}\left[ \frac{p_\theta(y \mid z, r)\, p(r)}{q_\psi(r \mid y)} \right] \tag{19}$$

$$\geq \mathbb{E}_{q_\psi(r\mid y)}\big[ \log p_\theta(y \mid z, r) \big] - D_{\mathrm{KL}}\big( q_\psi(r \mid y) \,\|\, p(r) \big) \quad \text{(Jensen)} \tag{20}$$

| Method | Params | PSNR ↑ | SSIM ↑ | LPIPS ↓ |
|---|---|---|---|---|
| CDC Wei et al. (2020) | 40.3M | 32.08 | 0.8585 | 0.3095 |
| SwinIR Liang et al. (2021a) | 11.8M | 31.74 | 0.8466 | 0.3317 |
| HAT Chen et al. (2023a) | 20.8M | 31.93 | 0.8506 | 0.3197 |
| KDSR Xia et al. (2022) | 18.8M | 31.97 | 0.8533 | 0.3116 |
| CDFormer Liu et al. (2024) | 25.0M | 31.77 | 0.8459 | 0.3320 |
| ResShift Yue et al. (2023) | 118.6M | 29.56 | 0.7812 | 0.3932 |
| Ours | 14.0M | **32.27** | **0.8608** | **0.2994** |

Table 3: Results on DRealSR test set ($\times 4$).

## A.2 Derivation of the Mutual-Information Identity

$$I(y; r; z) = I(r; z) - I(r; z \mid y) = I(r; y) - I(r; y \mid z) = I(z; y) - I(z; y \mid r) \quad \text{(interaction info)} \tag{21}$$

$$I(y; z, r) = I(y; z) + I(y; r \mid z) = I(y; r) + I(y; z \mid r) \quad \text{(chain rule)} \tag{22}$$

$$I(r; z) = I(r; y) + I(z; y) - I(y; z, r) + I(r; z \mid y) \quad \text{(rearrange)} \tag{23}$$

Under $(r, z, y) \sim p_{\text{data}}(x, y) \, q_\phi(r \mid y)$ with $z = D(x)$,

$$p(r, z \mid y) = q_\phi(r \mid y) \, p_{\text{data}}(z \mid y) \;\Rightarrow\; r \perp\!\!\!\perp z \mid y \;\Rightarrow\; I(r; z \mid y) = 0. \quad \text{(CI)} \tag{24}$$

Hence

$$I(r; z) = I(r; y) + I(z; y) - I(y; z, r). \tag{25}$$

(Here $z = D(x)$ is deterministic with fixed $D$, so $I(z; y)$ is constant w.r.t. learnable parameters.) (26)

## A.3 Additional Experimental Results

We train our method on the large-scale paired real-world dataset DRealSR Wei et al. (2020). We evaluate our model on DRealSR test set under scale of $\times 4$. All comparison models are retrained on the DRealSR training set with the same setting. The results are summarized in Table. 3. Our method outperforms state-of-the-art real-world SR methods and degradation representation approaches.

To further validate the discriminability of our degradation representation in modeling spatially variant real-world degradations, we additionally conduct super-resolution experiments on synthetic spatially variant blur kernels. On the BSD100 Martin et al. (2001) dataset, each image is degraded with spatially varying kernels, where different regions (patches) are subjected to different blur patterns according to a fixed degradation protocol proposed in MANet Liang et al. (2021b). As shown in Table. 4, although our representation does not utilize any GT kernel supervision, it achieves comparable or even superior guidance performance to supervised predicted kernels.

## A.4 Additional Experimental Details

**Training objective.** We optimize a pixelwise $\ell_1$ reconstruction loss together with a KL regularizer in SAVL, in line with the composed objective:

$$\mathcal{J}_{\text{SAVL}}(\theta, \psi) = -\mathbb{E}_{q_\psi(r \mid y)}\big[\log p_\theta(y \mid z, r)\big] + (1 + \lambda) \, D_{\text{KL}}\big(q_\psi(r \mid y) \,\|\, p(r)\big), \tag{27}$$

where the reconstruction and KL terms are estimated by the same amortized networks and neighborhoods as in SAVL-LM. In practice we model the per-pixel conditional likelihood as Laplace, so the reconstruction reduces to an $\ell_1$ loss:

$$\mathcal{L}_{\text{rec}} = \sum_{u \in \Omega} \big\| y(u) - \hat{y}_\theta(u; z, r) \big\|_1. \tag{28}$$

We train with tunable weights and KL annealing to prevent collapse:

$$\min_{\theta, \psi} \mathcal{L}_{\text{train}} = \alpha \, \mathcal{L}_{\text{rec}} + \beta_t \, D_{\text{KL}}\big(q_\psi(r \mid y) \,\|\, p(r)\big), \tag{29}$$

$$\beta_t = \beta_{\text{final}} \cdot \min\Big(1, \, \tfrac{t}{T_{\text{anneal}}}\Big), \quad \beta_{\text{final}} = 5 \times 10^{-4}, \quad T_{\text{anneal}} = 50{,}000 \text{ iterations}, \quad \alpha = 1. \tag{30}$$

| Method | Spatially Variant Kernel Type | | | | |
|---|---|---|---|---|---|
| | $\sigma_1 = a+b$ $\sigma_2 = ax+b$ $\theta = 0$ | $\sigma_1 = ay+b$ $\sigma_2 = ax+b$ $\theta = 0$ | $\sigma_1 = a+b$ $\sigma_2 = b$ $\theta = \pi x$ | $\sigma_1 = ay+b$ $\sigma_2 = ax+b$ $\theta = \pi x$ | $\sigma_1 \sim \mathcal{U}(b, a+b)$ $\sigma_2 \sim \mathcal{U}(b, a+b)$ $\theta \sim \mathcal{U}(0, \pi)$ |
| **Global Degradation Learning Method** | | | | | |
| RRDB-SFTWang et al. (2018b) + CDP Liu et al. (2024) | 24.40 / 0.5911 | 24.71 / 0.6003 | 24.78 / 0.6130 | 24.99 / 0.6176 | 24.91 / 0.6026 |
| **GT-Kernel Supervised Methods** | | | | | |
| RRDB-SFT + IKC Gu et al. (2019) | 24.64 / 0.5950 | 24.94 / 0.6162 | 24.81 / 0.6175 | 25.01 / 0.6174 | 24.95 / 0.6078 |
| RRDB-SFT + MANet Liang et al. (2021b) | **24.89 / 0.6030** | 25.21 / 0.6192 | 25.11 / 0.6197 | **25.24 / 0.6200** | 25.05 / 0.6118 |
| **Ours (No Kernel Supervision)** | | | | | |
| RRDB-SFT + Ours | 24.85 / 0.6018 | **25.26 / 0.6200** | **25.18 / 0.6199** | 25.22 / 0.6192 | **25.13 / 0.6130** |
| **Upper Bound** | | | | | |
| RRDB-SFTWang et al. (2018b) + GT | 24.98 / 0.6082 | 25.32 / 0.6255 | 25.33 / 0.6292 | 25.34 / 0.6264 | 25.30 / 0.6233 |

Table 4: Quantitative comparison on BSD100 dataset for spatially variant blur kernel degradation at scale ×4 and noise level 15. All results are reported in terms of PSNR/SSIM, evaluated on the Y channel.

**Model overview.** The network consists of a MANet encoder and a decoder that acts as a posterior estimator. The encoder takes the lr image $y$ and predicts a per-pixel diagonal Gaussian over the degradation code $r$; the decoder conditions on $r$ and the hr image $x$ to produce $\hat{y}$.

**Encoder.** We use MANet Liang et al. (2021b) as the backbone (Conv head $\rightarrow$ MAConv-based residual blocks with shared local receptive fields $\rightarrow$ Conv tail), with GroupNorm (32 groups) and SiLU/Swish activations. The **degradation representation dimension is** 64, so the encoder outputs $2 \times 64$ channels (mean and log-variance) for a diagonal Gaussian $q_\psi(r \mid \cdot) = \mathcal{N}(\mu, \mathrm{diag}\,\sigma^2)$, and sampling uses the standard reparameterization $r = \mu + \sigma \odot \varepsilon$.

**Decoder (posterior estimator).** The decoder fuses the degradation code with image-conditioned features to reconstruct $\hat{y}$. A single fixed Gaussian blur (kernel $9 \times 9$, $\sigma = 9.0$) forms a low-frequency base and a complementary high-frequency condition; features are fused via an SFT-style block and processed by an RRDB Wang et al. (2018b) trunk (default 8 blocks, growth channels 32, residual scale 0.2), followed by a 3-channel prediction head.

**Key hyperparameters.** Unless otherwise specified: base channels 32; channel multipliers $[1, 1, 2, 4]$; encoder code dimension 64; decoder encoding scale 0.25; GroupNorm(32); truncated normal weight init.

## A.5 Further Qualitative and Quantitative Evidence for Spatial Modeling and Degradation–Content Decoupling

In this section, we provide additional experimental evidence, both qualitative and quantitative, demonstrating that SAVL (1) effectively models spatially varying degradations and (2) successfully decouples degradation from content. This section includes the t-SNE analysis, the unsupervised classification study, and more ablation experiments.

### A.5.1 Modeling the Spatially Variant Degradations

To further illustrate the spatially variant property in a more controlled and quantitative manner, we divide a high-resolution image from DRealSR into patches and randomly apply different degradations to different patches. We then extract pixel-level feature representations and assign each pixel a label corresponding to its degradation type. Using t-SNE, we visualize these representations and perform unsupervised k-means clustering to assess how well the pixel degradations can be separated. A strong representation should exhibit clear clustering patterns and achieve high clustering accuracy.

As shown in Fig. 9, we adopt three different strategies to synthesize spatially varying degradation types.

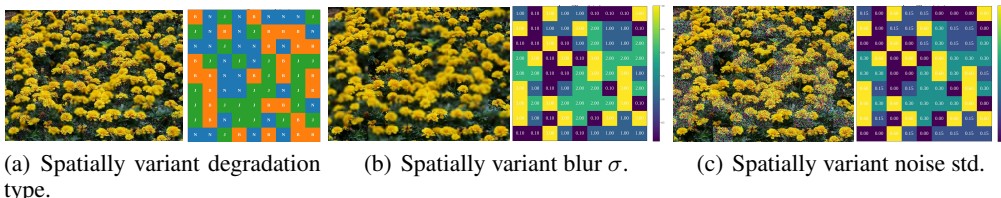

(a) Spatially variant degradation type.
(b) Spatially variant blur $\sigma$.
(c) Spatially variant noise std.

Figure 9: Different spatially varying degradation strategies and their degradation map.

| Spatially Variant Degradation | | Accuracy | | | |
|---|---|---|---|---|---|
| Category | Setting | Ours | MANet | Ours w/o SAVL | Ours w/o SAVL + CLUB |
| Variant Type | Noise / Blur / JPEG | **72.44**% | 43.66% | 48.02% | 41.68% |
| Variant Blur Level | $\sigma$ = 0.1 / 1.0 / 2.0 / 3.0 | **61.20**% | 37.32% | 39.34% | 32.48% |
| Variant Noise Strength | std = 0 / 0.15 / 0.30 / 0.60 | **57.60**% | 37.78% | 51.36% | 37.64% |

Table 5: Clustering accuracy under different spatially varying degradation settings. Our method achieves higher clustering accuracy under 3 spatially variant degradation settings, indicating a stronger ability to distinguish spatially varying degradations.

1. Each patch is randomly assigned one of three degradations: noise, blur, or JPEG compression.
2. Each patch is randomly assigned a blur kernel with a different $\sigma$ value.
3. Each patch is randomly assigned Gaussian noise with a different standard deviation.

Our method is compared against MANet Liang et al. (2021b), as other approaches model only global degradation and lack pixel-level representation capability. We also include ablation variants for completeness. The results are presented in Fig 10 and Table 5.

We then provide more visual evidence demonstrating SAVL's effectiveness in modeling spatially varying degradations. Because the deterministic variant lacks learnable variance, we statistically examine the local variance of both deterministic representations as well as the predicted mode of our per-pixel representation. As shown in Fig 11, the deterministic representations exhibit limited spatial variability—their spatial variance is relatively uniform and is influenced primarily by image brightness (RGB values) rather than by actual degradation factors.

### A.5.2 MITIGATING THE DEGRADATION-CONTENT ENTANGLEMENT.

Compared to spatial variability, the model's ability to decouple degradation from content is more challenging to visualize directly. Beyond the t-SNE plot presented in the main text, we further include a visualization based on a proxy experiment. In this experiment, we extract the content features of a clean image (image A) and the degradation representation of a degraded image (image B), and feed them jointly into the decoder. If degradation and content are truly decoupled, the decoded result should preserve the content of image A while exhibiting the degradation characteristics of image B. Crucially, the hybrid image should contain no content or texture originating from image B. As shown in Fig 12, removing SAVL causes the decoded output to be dominated entirely by the content of the degraded image B; the CLUB constraint alleviates this issue to some extent, but it still introduces noticeable edges and textures from image B. Only with SAVL do we obtain a clean hybrid result that matches the intended combination of "content A + degradation B." This demonstrates that our degradation embedding is a purely "degradation" representation decoupled from the content.

### A.6 FURTHER ANALYSIS OF THE POSTERIOR MEAN (THE "MODE")

In the main text, we describe the posterior mean of our representation as encoding the type of degradation. Here, we provide additional experimental analysis to further clarify this interpretation. By fixing all but one degradation factor and varying the remaining one, we examine how individual

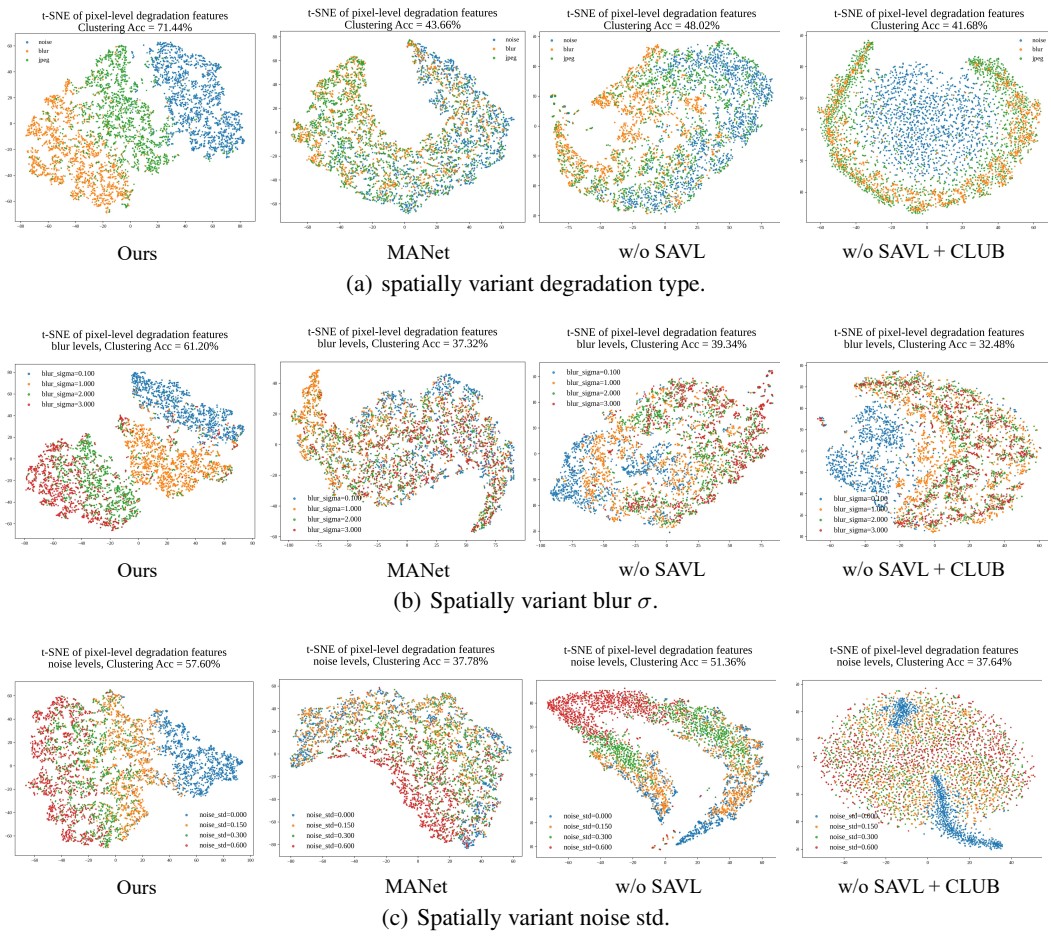

(a) spatially variant degradation type.

(b) Spatially variant blur $\sigma$.

(c) Spatially variant noise std.

Figure 10: The t-SNE visualization and clustering accuracy of the pixel-level representations show that our features naturally cluster according to the same degradation type or parameter. Moreover, our method achieves higher clustering accuracy, indicating a stronger ability to distinguish spatially varying degradations.

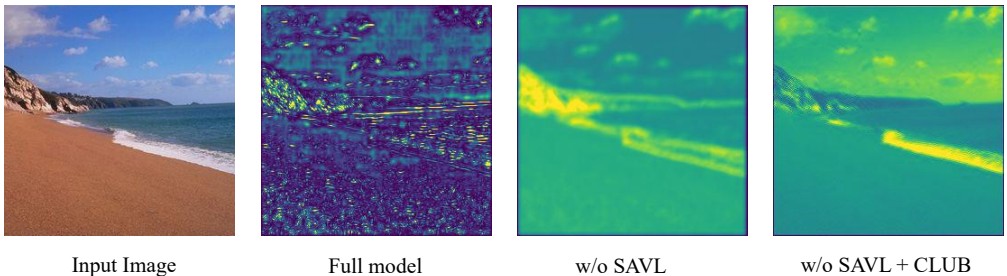

| Input Image | Full model | w/o SAVL | w/o SAVL + CLUB |

Figure 11: The deterministic representations exhibit limited spatial variability—their spatial variance is relatively uniform and is influenced primarily by image brightness (RGB values) rather than by actual degradation factors.

dimensions of the mean vector correspond to specific degradation changes. We observe consistent and predictable variations along certain dimensions of the mean vector as the associated degradation factor is altered.

Concretely, we conduct this experiment by fixing two degradation parameters (e.g., noise and JPEG) while systematically varying the third (e.g., blur sigma from 0.2 to 4.0). At each step, we extract the

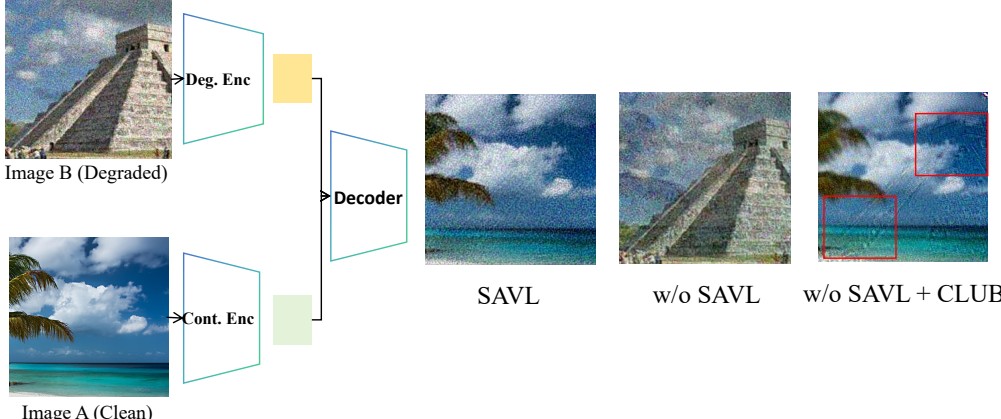

Figure 12: Removing SAVL makes the decoded output dominated by the content of the degraded image B, while the CLUB constraint only partially improves the result and still introduces edges and textures from image B. Only SAVL produces a clean hybrid image reflecting "content A + degradation B," demonstrating that our degradation embedding captures degradation alone and remains decoupled from content.

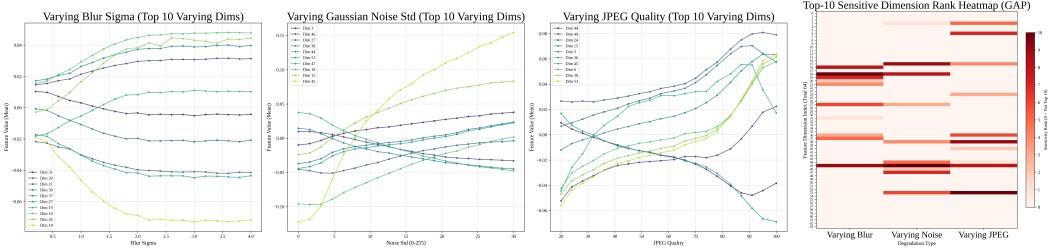

(a) When a particular degradation factor changes, certain dimensions of the mean vector exhibit monotonic and predictable trajectories.

(b) Subspaces of different degradation factors

Figure 13: The posterior mean changes smoothly and predictably along specific dimensions as a single degradation factor varies, illustrating a stable and coherent mapping between degradation strength and the corresponding mean-vector dimensions.

64-D posterior mean vector (via global average pooling). We identify the top ten dimensions with the largest variance under each perturbation and treat these as the subspace most representative of that particular degradation.

As shown in Fig. 13, our analysis reveals two key findings:

**(1)Consistent change.** The dimensions with the highest variance exhibit smooth, monotonic trends as the degradation strength increases, indicating a predictable and well-behaved mapping.

**(2) Disentanglement.** The dimensions most sensitive to blur tend to differ from those responsive to noise or JPEG, suggesting that the encoder organizes different degradation factors into partially distinct subspaces.

### A.7 FURTHER ANALYSIS ON OOD DEGRADATIONS

To investigate the model's behavior on entirely out-of-distribution degradation types that are absent during training (e.g., motion blur, atmospheric haze, or sensor-specific noise patterns), we synthesize motion blur with varying kernel lengths and haze with different concentrations. These settings allow us to assess the robustness of our learned degradation representation under truly novel degradation conditions.

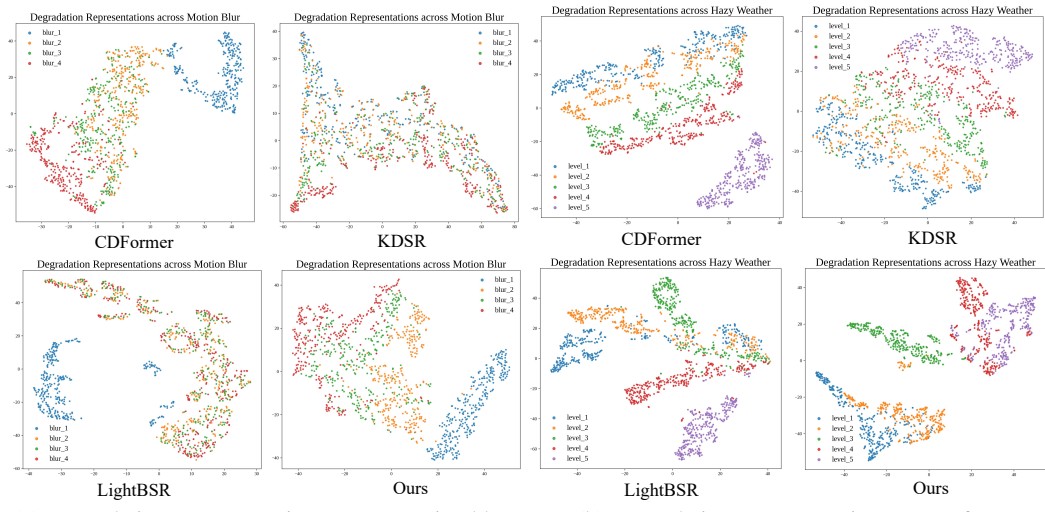

(a) Degradation Representations across motion blur.    (b) Degradation Representations across fog.

Figure 14: Visualization of the learned representation space. Our representation also exhibits stronger discriminability on the OOD data.

**Sensor-specific noise patterns have already been examined in the main text through the SVSR dataset Aakerberg et al. (2024), which contains noise characteristics unique to specific camera sensors, and our training set does not include any SVSR data. Please refer to fig 6.**

We further use t-SNE to analyze the distribution of the degradation representations in the feature space. As shown in Fig. 14, the probabilistic nature of SAVL provides enhanced robustness to domain shifts compared to deterministic methods.

### A.8 STABILITY, CONVERGENCE, AND TRAINING COST

Our training objective consists primarily of the KL term and the reconstruction loss. To prevent posterior collapse, we apply KL annealing, gradually increasing the KL weight from 0 to 5e-4 over the first 50k iterations. As shown in Fig. 15, the TensorBoard curves indicate stable training dynamics: the KL loss and reconstruction loss counterbalance each other and remain stable once annealing completes, while the reconstruction loss steadily decreases and converges at around 200k iterations. These observations suggest that the mutual-information constraints remain well-behaved during optimization, and the model's capacity to encode degradations improves consistently throughout training. Our approach exhibits strong training stability and reliable convergence throughout the learning process.

Our training time is approximately 48 hours for learning the degradation representation and another 48 hours for training the super-resolution network. This is comparable to multi-stage implicit degradation–representation methods such as CDFormer Liu et al. (2024) and KDSR Xia et al. (2022), which also require two-stage training involving separate teacher and student models. Our training cost is somewhat higher than lighter-weight implicit approaches like DASR and LightBSR, but remains within a similar overall scale.

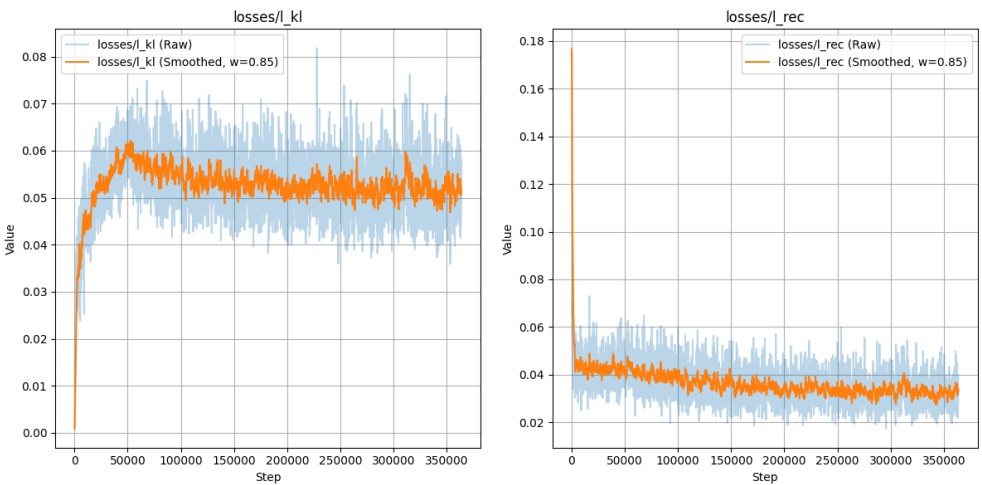

Figure 15: TensorBoard loss curves of our training process.

