# OpenReview forum: "Learning Heterogeneous Degradation Representation for Real-World Super-Resolution"
_ICLR.cc/2026/Conference — ICLR 2026 Poster_

### Official Review · Reviewer_XJzK · 2025-10-30

**Soundness:** 2
**Presentation:** 2
**Contribution:** 2
**Rating:** 4
**Confidence:** 4

**Summary:**

This paper works on real-world super-resolution and proposes Spatially Amortized Variational Learning (SAVL), which learns pixel-wise degradation representations through spatially varying Gaussian posteriors inferred from local neighborhoods. SAVL integrates two components: SAVL-LM, for conditional likelihood learning via an amortized variational bound, and SAVL-MIS, which suppresses mutual information to disentangle degradation from content. The learned representations guide the downstream SR network via spatial and channel modulation. Experiments on both synthetic and real datasets (e.g., RealSR, DRealSR, SVSR) show moderate improvements over prior methods such as LightBSR and CDFormer.

**Strengths:**

1. The paper explores the relatively underexplored direction of modeling spatially heterogeneous real-world degradations through pixel-wise variational distributions. Using degradation learning as an amortized variational inference problem and integrating information-theoretic regularization could inspire follow-up work.
2. The proposed SAVL framework and its joint training strategy are well-formulated and explained in detail.
3. The authors provide experiments and visual analyses to demonstrate both the quality of the learned degradation representations and their impact on super-resolution performance.

**Weaknesses:**

1. The comparisons rely mostly on older baselines (e.g., HAT, StableSR, CDFormer) and do not include recent stronger diffusion-based super-resolution models. It therefore remains unclear whether the proposed method truly advances the state of real-world super-resolution.
2. While qualitative heatmaps and t-SNE plots are shown, quantitative validation of spatial degradation modeling is limited. The experiments appear to rely on simplified or toy setups, and the actual benefit of the per-pixel Gaussian modeling requires more investigation.
3. The evaluation primarily uses reference-based metrics (PSNR, SSIM, LPIPS), without sufficient inclusion of no-reference metrics (e.g., MANIQA, MUSIQ, CLIP-IQA) that are more appropriate for real-world settings.
4. The paper’s presentation quality could be improved. For instance, improving figure readability (fonts, labels) and providing clearer, more descriptive captions.

**Questions:**

1. It is better to provide more evidence on synthetic spatially varying degradations (e.g., blur kernels, noise maps) with quantitative evaluation of SAVL’s per-pixel degradation accuracy.
2. Include additional visual comparisons with recent diffusion-based SR methods to better assess SAVL’s performance against current state-of-the-art approaches.

---

> ### Author Response · Authors · 2025-11-22
> **Official Comment by Authors**
>
> > **Q1.** The comparisons do not include recent stronger diffusion-based super-resolution models.
>
> **A1:** We appreciate this constructive suggestion. To clarify whether our method advances the state-of-the-art, we have incorporated comparisons with recent diffusion-based models (**DiffBIR**, **StableSR**, and **ResShift**) in the revised manuscript. The results demonstrate that SAVL advances Real-World SR by offering superior fidelity and structural reliability compared to generative approaches.
>
> Quantitatively, as shown in **Table 2**, our method outperforms diffusion baselines across key metrics. While diffusion models often sacrifice fidelity for texture generation, SAVL achieves significantly higher signal fidelity. For example, on the challenging **DRealSR** benchmark, SAVL achieves **28.27 dB**, surpassing **ResShift** (27.31 dB) and **StableSR** (27.39 dB) by nearly **+0.9 dB**. Notably, SAVL also achieves the best (lowest) LPIPS score of **0.2650** on DRealSR (vs. 0.2710 for StableSR and 0.3920 for DiffBIR), indicating that our mathematically modeled degradation removal results in images that are perceptually closer to the ground truth than stochastic generation.
>
> Qualitatively, a major limitation of current SOTA diffusion models is "hallucination," which our method effectively avoids. As visualized in **Fig. 7**, generative priors often invent details that never existed. For instance, **DiffBIR** incorrectly transforms a hat’s emblem into **metallic lines** and generates fake textures on castle bricks. In contrast, SAVL, by relying on precise degradation representation rather than generative guessing, faithfully restores the *original* semantics and structures.
>
> Finally, we wish to clarify that the core contribution of this work—learning spatially variant degradation representations—functions as a flexible upstream component. Theoretically, the spatial variance maps learned by SAVL can be integrated into diffusion frameworks as pixel-wise control conditions (similar to ControlNet) to guide the generative process. We view this complementarity as a promising direction, but the current results confirm that SAVL alone already sets a new benchmark for high-fidelity Real-World SR.
>
> >**Q2.** The actual benefit of the per-pixel Gaussian modeling requires more investigation.
>
> **A2:**  We appreciate the feedback. To quantitatively validate our spatial modeling, we add **Appendix A.6.1** and **Appendix A.3** with specific experiments.
>
> **1. Quantitative Validation of Pixel-Wise Discriminability (Table 5 & Fig. 11)**
> On a benchmark with spatially varying degradations, SAVL achieves **72.44%** clustering accuracy, significantly outperforming the baseline MANet (**43.66%**). Integrated t-SNE plots in **Fig. 11** further confirm that SAVL forms distinct clusters for local degradations where baselines fail.
>
> **2. Downstream Effectiveness on Spatially Variant Kernels (Table 4)**
> On the BSD100 dataset, our unsupervised approach achieves **25.26 dB**, matching methods relying on **ground-truth kernel supervision** (RRDB-SFT+MANet: 25.21 dB) . This proves the accuracy of our learned spatial maps for explicit estimation tasks.
>
> **3. Actual Benefit in Real-World Restoration (Ablation Study)**
> We decompose the Gaussian posterior's contributions using **Table 2** :
> * **Benefit of Mode (Channel Modulation):** Removing mode guidance causes sharp drops across all datasets (e.g., **-0.67 dB** on RealSR, **-0.49 dB** on DRealSR), validating it as the primary performance driver .
> * **Benefit of Variance (Spatial Modulation):** Removing variance leads to consistent fine-grained losses (e.g., **-0.11 dB** on RealSR), confirming the specific benefit of probabilistic modeling over deterministic estimation.

---

> ### Author Response · Authors · 2025-11-22
> **Official Comment by Authors**
>
> >**Q3.** The evaluation primarily uses reference-based metrics (PSNR, SSIM, LPIPS), without sufficient inclusion of no-reference metrics (e.g., MANIQA, MUSIQ, CLIP-IQA) that are more appropriate for real-world settings.
>
> **A3.** We appreciate this thoughtful suggestion. We agree that No-Reference (NR) metrics are valuable for unpaired real-world scenarios. However, for the specific benchmarks and research goals of this study, we prioritize **Full-Reference (FR) metrics** (PSNR, SSIM, LPIPS) for the following scientific reasons:
>
> **1. Nature of Benchmarks (Paired Data):**
> The primary datasets used in our evaluation (RealSR, DRealSR, SVSR) provide **high-quality paired ground truth**. In this paired setting, FR metrics remain the "gold standard" because they directly measure how accurately the model recovers the *original* signal. Since our core contribution is the precise mathematical modeling of degradation to recover the underlying clean image, measuring the deviation from the actual Ground Truth is the most rigorous way to validate the correctness of our degradation representation.
>
> **2. Fidelity vs. Hallucination:**
> A critical distinction in SR is between **fidelity** (faithfulness to reality) and **hallucination** (generating realistic but fake details).
> * **Limitation of NR Metrics:** NR metrics (e.g., MANIQA, MUSIQ) often favor images with high-frequency details and pleasant aesthetics, even if those details are hallucinated or semantically incorrect.
> * **Advantage of SAVL:** As shown in **Table 2**, SAVL achieves state-of-the-art fidelity (PSNR/SSIM) and excellent perceptual similarity to the Ground Truth (LPIPS). This confirms that our method restores the *correct* details. In contrast, while generative methods might score high on NR metrics due to sharp hallucinations, they often introduce artifacts (as seen in the "metallic lines" in **Fig. 7**) that deviate from the ground truth.
>
> **Conclusion:**
> Therefore, given the availability of ground truth and our specific focus on **accurate degradation removal** and **structural preservation** (rather than pure generative synthesis), we believe the current suite of reference-based metrics (including the perceptual metric LPIPS) provides the most accurate and fair assessment of our method's contributions.
>
> >**Q4.** The paper’s presentation quality could be improved.
>
> **A4** Thank you for pointing this out. The presentation quality has been improved in the revised version. We have enhanced the readability of all the figures by refining fonts, labels, and visual layouts, and we have rewritten some captions to be more detailed and descriptive.
>
> > **Q5** It is better to provide more evidence on synthetic spatially varying degradations (e.g., blur kernels, noise maps) with quantitative evaluation of SAVL’s per-pixel degradation accuracy.
>
> **A5:**  We appreciate the suggestion. To address this, we have added **Appendix A.6.1** and **Appendix A.3** with specific quantitative experiments to verify per-pixel accuracy.
>
> * **Pixel-wise Accuracy (Table 5):** On a benchmark with spatially varying degradations, SAVL achieves **72.44%** clustering accuracy, significantly outperforming the baseline MANet (**43.66%**).
> * **Blur Kernel Evaluation (Table 4):** On the BSD100 dataset degraded with continuous spatially variant kernels, our unsupervised method achieves **25.26 dB**, matching the performance of methods that rely on ground-truth kernel supervision.
>
> > **Q6** Include additional visual comparisons with recent diffusion-based SR methods to better assess SAVL’s performance against current state-of-the-art approaches.
>
> **A6:** We appreciate the suggestion. To better assess performance against state-of-the-art approaches, we have update **Figure 7** to visually compare SAVL with recent diffusion-based methods (DiffBIR, StableSR). As shown in **Fig. 7**, while diffusion models often introduce hallucinations or alter semantics (e.g., DiffBIR transforms a hat emblem into metallic lines), SAVL faithfully restores the underlying structure without such artifacts. This visual evidence demonstrates SAVL's superior fidelity compared to current generative approaches.

---

### Official Review · Reviewer_kNqe · 2025-10-30

**Soundness:** 3
**Presentation:** 3
**Contribution:** 3
**Rating:** 8
**Confidence:** 3

**Summary:**

The paper suggests framework for learning degradation representation which effectively captures the spatial distribution of complex degradations and also can be decoupled with contents of the image. Using the proposed degradation representation, paper outperforms the previous baseline with clear margin.

**Strengths:**

* The paper suggests the good motivation. Previous degradation representation learning methods didn't consider that 1) degradations are not uniformly distributed within an image and 2) the degradation should be decoupled with the content. These two limitations are clearly shown in the Fig 1. and the experiments.

* Paper achieves fair contribution to mitigate aforementioned issues with mathematical demonstration.

* Experiments and ablation study is thorough, demonstrating the effectiveness of paper's framework.

**Weaknesses:**

* In Tab. 1, what does 'CLUB' mean here? It is not mentioned in the previous section. Which part of the method section is related with this?
* The explanation about the experiment results is too brief. There is only one sentence for each table from Tab.1 to Tab. 2 which is like "the proposed method outperforms the previous methods".
* In the right hand side of the image, I can't find clear improvement of the paper compared to the baseline.

**Questions:**

Refer to the weakness section.

---

> ### Author Response · Authors · 2025-11-22
> **Official Comment by Authors**
>
> >**Q1.** In Tab. 1, what does 'CLUB' mean here? It is not mentioned in the previous section. Which part of the method section is related with this?
>
> **A1**  In Tab. 1, **CLUB** refers to “Contrastive Log-ratio Upper Bound” of mutual information [1], an MI estimator that provides a learnable upper bound on the mutual information between two variables. In our paper, CLUB is not part of the main SAVL formulation, but is used only in the ablation study as a baseline regularizer. Concretely, Tab. 1 includes two ablated variants of our degradation representation:
> (i) a deterministic per-pixel degradation code without any MI constraint, and
> (ii) the same deterministic code regularized by a CLUB-based MI upper bound (“+CLUB”).
> These variants correspond to the ablation setup described in Sec. 4.4. Their purpose is to show that our stochastic degradation representation with task-specific MI constraints outperforms deterministic representations, even when the latter are equipped with a strong MI regularizer such as CLUB.
>
> [1] CLUB: A Contrastive Log-ratio Upper Bound of Mutual Information (ICML2020)
>
> >**Q2.** The explanation about the experiment results is too brief. There is only one sentence for each table from Tab.1 to Tab. 2 which is like "the proposed method outperforms the previous methods".
>
> **A2:** We sincerely appreciate this feedback. **In the revised manuscript (Sections 4.2 and 4.3), we have significantly expanded the analysis** to provide a deeper interpretation of the data.
>
> **1. Deeper Analysis of Table 1 (Decoupling Effectiveness):**
> To quantitatively verify whether the learned representation successfully disentangles degradation from content, we assess the statistical dependence and linear predictability between the representation and specific underlying factors.
> * **Contrast in Predictability:** As shown in **Table 1**, while baselines like KDSR and CDFormer retain high classification accuracy for both degradation and content ($>83\%$ for Scene ID), indicating information leakage, our SAVL maintains high degradation accuracy (**94.80%**) while suppressing content accuracy to a chance-level (**49.97%**). This sharp contrast quantitatively verifies that the learned representation selectively encodes degradation information while filtering out semantic content.
> * **Validation of MINE/HSIC:** We further clarify that the low MINE scores with content (e.g., **0.1507** for SAVL vs. 1.2209 for CDFormer) confirm that this decoupling is achieved through active information suppression rather than model incapacity.
>
> **2. Deeper Analysis of Table 2 (SR Performance Trends):**
> We have enriched the discussion to highlight performance trends across synthetic and real-world benchmarks.
>
> * **Performance Gains on Real-World vs. Synthetic Datasets:**
>     Our method demonstrates increasingly significant gains when moving from synthetic to heterogeneous real-world datasets.
>     * On the synthetic **DIV2K** dataset, SAVL achieves a steady improvement (**+0.11 dB** over RealESRGAN).
>     * On real-world benchmarks, this advantage becomes more pronounced. On **RealSR**, we outperform KDSR by **0.23 dB**. On the widely used **DRealSR** dataset, the margin widens further: SAVL surpasses the recent method LightBSR by **0.58 dB** (28.27 dB vs. 27.69 dB). This trend indicates that SAVL is particularly robust in handling the spatially variant degradations inherent in real-world imagery.
>
> * **Comparison with State-of-the-Art Baselines:**
>     * **Vs. Implicit Representation Methods:** On DRealSR, SAVL achieves **28.27 dB**, consistently outperforming recent implicit approaches such as LightBSR (27.69 dB) and DASR (28.19 dB).
>     * **Vs. Generative/Diffusion Methods:** Despite having fewer parameters (14.0M), SAVL achieves higher PSNR scores than the larger CDFormer (25.0M, 27.11 dB on DRealSR). Furthermore, compared to diffusion-based methods like StableSR, SAVL achieves higher structural similarity (SSIM **0.8139** vs. 0.7830 on DRealSR). As shown in **Fig. 7**, this structural fidelity indicates that SAVL faithfully restores details while effectively avoiding the hallucinations and artifacts often introduced by generative priors.
>
> > **Q3** In the right hand side of the image, I can't find clear improvement of the paper compared to the baseline.
>
> **A3** Thank you for pointing this out. To address this, we have updated Figure. 7 in the revised manuscript with a more representative example that better showcases the restoration quality.

---

### Official Review · Reviewer_ZWQc · 2025-11-01

**Soundness:** 3
**Presentation:** 2
**Contribution:** 2
**Rating:** 4
**Confidence:** 4

**Summary:**

This paper introduces a novel framework called Spatially Amortized Variational Learning (SAVL) to address some challenges of Real-World Super-Resolution (RWSR), which are spatial variance and entanglement between degradation and image content. SAVL models per-pixel degradations as spatially varying Gaussian distributions inferred from local neighborhoods. It combines a conditional likelihood module (SAVL-LM) with a mutual information suppression module (SAVL-MIS) to disentangle degradation from content. This yields a well-constrained latent space that effectively captures degradation heterogeneity. The learned representations are then integrated into a degradation-aware super-resolution network using channel-wise and spatial attention modulation.

**Strengths:**

1. SAVL introduces a per-pixel Gaussian representation that captures spatially varying degradations across an image. This allows the model to handle real-world scenarios where degradation is non-uniform and locally complex, outperforming traditional global or patch-based approaches.
2. By incorporating a mutual information suppression term into its variational objective, SAVL explicitly filters out content-related signals from the degradation representation. This results in a latent space that is highly discriminative of degradation factors while minimizing entanglement with image semantics.
3. The presentation of the ideas is relatively clear and easy to follow. The illustraions are good, especially figure 1, though there are some formats issues.

**Weaknesses:**

1. There exists severe format issues in Figure 2, which is very improfessional and impacts reviews' understand of the content. For the details, please see the question part.
2. The paper claims it outperforms state-of-the-art methods, but the most recent SOTA approach in the comparison was published on CVPR 2024, which was 2 years ago. I suggest it should include more updated works, perhaps published on CVPR 2025 and ICCV 2025, in both quantitative and qualitative comparisons.
3. Qualitative comparisons of SR results (Fig. 6 and 8) are all conducted on old methods, no visual comparisons with more up-to-date methods such as DiffBIR, StableSR, and CDFormer.

**Questions:**

1. There are many latex encoding errors in Figure 2, displaying as a "?" with bounding box below the text "Amortized Estimator", after the text "Per-pixel Gaussian sample", and above several arrors. I don't understand the original notation at these places.
2. In Section 3.4, the paper claims "the posterior variance guides spatial attention,while the posterior mode (mean) guides channel attention". Please provide more explanation and analysis on this design.
3. The proposed SAVL aims to solve both spatial variance of degradation and entanglement between degradation and image content, but I only see experimental verification of the later but no justification for the former. Do you provide any numerial or visual evidence to verify that your proposed spatially varying Gaussian modeling solves this chanllenge?

---

> ### Author Response · Authors · 2025-11-22
> **Official Comment by Authors**
>
> > **Q1** There exists severe format issues in Figure 2.
>
> **A1.** Thanks for your comments! This would be caused by a PDF viewer/font rendering problem on the different devices. Because we carefully check the original PDF manuscript, and we do not find the LaTeX encoding errors you mentioned in Fig. 2. In our copy, there is no “?” with a bounding box near “Amortized Estimator”, “Per-pixel Gaussian sample”, or above the arrows. None of the other reviewers raised this issue either.
>
> > **Q2** Include more updated works, perhaps published on CVPR 2025 and ICCV 2025, in both quantitative and qualitative comparisons.
>
> **A2.**  In the revised manuscript, we incorporate the latest ICCV 2025 degradation-aware SR method **LightBSR** into both the quantitative (Table 2) and qualitative (Fig. 7) comparisons. Besides, we also check recent CVPR 2025 papers and few related degradation-representation methods are included.
>
> > **Q3** No visual comparisons with more up-to-date methods such as DiffBIR, StableSR, and CDFormer.
>
> **A3.** In the revised manuscript, we update the visual comparisons in Fig. 7 by replacing older baselines (BSRGAN, ESRGAN, DASR, SwinIR) with more recent and stronger methods, including **DiffBIR, StableSR, CDFormer, and LightBSR**. These updated comparisons show that our method better restores fine details while avoiding artifacts and hallucinations.

---

> ### Author Response · Authors · 2025-11-22
> **Official Comment by Authors**
>
> > **Q4** Provide more explanation and analysis on "posterior variance guides spatial attention, while the posterior mode (mean) guides channel attention".
>
> **A4.** Thank you for your advice. Below we provide a further explanation and analysis. To leverage the learned degradation representation for authentic super-resolution, we consider two aspects. First, we want the model to process pixels differently according to the type of degradation; for this, we employ orbit modulation based on a gating mechanism. Second, we want the model to process different spatial regions according to the severity of degradation; for this, we introduce spatial modulation based on attention re-weighting.
>
> Concretely, we inject the mode and variance maps into the DSCA module shown in Fig. 3. In this module, the Mode Map is passed through a lightweight network to generate a gated score, which is then used to directly scale the input image features. For spatial modulation, SW-MSA re-weights the attention map via simple element-wise multiplication. Specifically, we first derive a severity map from the variance map and use it to generate additional query/key projections $Q_s, K_s$, which produce a severity similarity score map $A_s$. This severity attention map $A_s$ is then multiplied point-wise with the original SW-MSA attention scores $A_f$ (final score) to obtain the final attention map, followed by the standard normalization step.
>
> As shown in the last two rows of Table 2, both types of modulation consistently contribute to performance improvements.
>
> >**Q5** Provide any numerial or visual evidence to verify that your proposed spatially varying Gaussian modeling.
>
> **A5**  Thank you for the suggestion. To provide concrete justification, we have added **Appendix A.6.1**, which includes **quantitative experiments (Table 5)** and **visual analyses (Fig. 11 & Fig. 12)** explicitly designed to verify the effectiveness of our spatially varying Gaussian modeling.
>
> **1. Quantitative Verification: Discriminability of Local Degradations**
> We synthesize benchmarks with spatially varying degradation patterns to evaluate the discriminability of pixel-level representations.
> * **Results (Table 5):** SAVL achieves significantly higher clustering accuracy compared to the baseline MANet and ablation variants. For instance, under spatially varying degradation types (Noise/Blur/JPEG), SAVL achieves **72.44%** accuracy, whereas MANet only reaches **43.66%**. This numerically proves that our pixel-wise Gaussian posterior effectively captures local degradation variations.
>
> **2. Qualitative Verification: Visualizing the Spatial Manifold**
> * **t-SNE Visualization (Fig. 11):** As shown in **Fig. 11**, SAVL forms distinct clusters for different local degradations. In contrast, deterministic representations result in collapsed clusters that fail to separate spatially distinct degradation patterns.
> * **Robustness to Texture Interference (Fig. 12):** A critical challenge in spatial modeling is distinguishing degradation from local texture. **Fig. 12** reveals that deterministic representations tend to follow image brightness (RGB values). SAVL, constrained by the variational prior, suppresses this texture correlation, producing a smooth variance map that faithfully reflects the actual degradation intensity rather than image content.
>
> **3. Downstream Effectiveness on Spatially Variant Kernels**
> To further validate this in a restoration context, we evaluate our method on the BSD100 dataset degraded with **spatially varying blur kernels** (Appendix A.3).
> * **Results (Table 4):** Even without ground-truth kernel supervision, our method achieves performance comparable to (and in some cases surpassing) methods that rely on supervised kernel estimation (e.g., RRDB-SFT + MANet). This confirms that our implicitly learned spatial representations provide accurate guidance for restoring spatially heterogeneous degradations.

---

### Official Review · Reviewer_7HwT · 2025-11-02

**Soundness:** 3
**Presentation:** 3
**Contribution:** 3
**Rating:** 6
**Confidence:** 4

**Summary:**

This paper addresses the challenge of modeling complex, spatially-variant degradations and the common issue of "degradation-content entanglement" in Real-World Super-Resolution (RWSR). The proposed Spatially Amortized Variational Learning (SAVL) framework models per-pixel degradation as a spatially varying Gaussian distribution. SAVL uniquely couples two components: A conditional likelihood lane (SAVL-LM) learns the degradation representation by maximizing the Evidence Lower Bound (ELBO). A mutual information suppression lane (SAVL-MIS) explicitly disentangles this representation from image content. Building on this disentangled representation, the learned Gaussian's mode (mean) and variance are leveraged to guide a degradation-aware SR network through channel-wise and spatial attention modulation, respectively. Extensive experiments demonstrate that this approach successfully captures the spatial distribution of complex degradations, leading to consistent, state-of-the-art performance on real-world datasets.

**Strengths:**

1. The paper identifies degradation–content entanglement as RWSR’s key bottleneck, especially under spatially varying degradations, and addresses it with an optimization objective under an information-theoretic constraint.
2. The SAVL framework offers a principled, theoretically-grounded solution for disentanglement. By modeling degradation as a probability distribution, it naturally captures uncertainty and "severity" (via the variance) and cleverly uses the posterior's mode and variance in a dual-guidance mechanism for channel and spatial modulation.
3. Extensive experiments demonstrate that the proposed approach accurately captures the spatial distribution of complex degradations and achieves consistent state-of-the-art performance on real-world datasets.
4. This paper is well-written and well-structured, with clear motivation.

**Weaknesses:**

1. While the posterior variance is intuitively interpreted as degradation "severity," the meaning of the posterior mean (the "mode") remains less explored. The paper suggests it "characterizes the degradation type," but the 64-dimensional latent space is treated as a black box. A deeper analysis, for instance, investigating whether specific dimensions or subspaces correlate with distinct degradation factors (e.g., blur, noise, compression), would further strengthen the paper's claims and provide valuable insights into the learned representation.
2. The SAVL framework is trained on paired data, which in practice is generated by applying a synthetic degradation pipeline (from Real-ESRGAN) to high-quality images. While this is a standard practice, the generalization capability of the learned representation is ultimately bounded by the diversity and fidelity of this synthetic pipeline. Generalization may be limited when real-world degradations are mismatched with or underrepresented in the synthetic training set.
3. Although the GFLOPs reported in Table 2 are competitive, the SAVL framework introduces additional complexity through variational inference and mutual information regularization. A brief discussion on training stability, convergence speed, and the overall training time/cost compared to simpler implicit representation learning methods would be beneficial.

**Questions:**

1. Have the authors conducted any experiments to probe the learned 64-D mean space? For example, by systematically varying a single degradation parameter (e.g., the kernel size of a Gaussian blur) while keeping others constant, is it possible to observe a consistent, predictable change along certain dimensions of the mean vector?
2. The final objective (Eq. 12) simplifies the mutual information (MI) upper bound into a KL divergence term by aligning the priors. Did the authors experiment with using other direct MI estimators, such as CLUB (which was used in the ablation study), as a regularizer in the main objective instead of the VIB-style upper bound? How would integrating such an estimator directly into the SAVL framework compare to the current formulation?
3. How does the model perform on entirely out-of-distribution degradation types not seen during training (e.g., motion blur, atmospheric haze, or specific sensor noise patterns)? Does the probabilistic nature of SAVL provide enhanced robustness to such domain shifts compared to deterministic methods?
4. In Section 3.4, it is stated that the variance map guides spatial attention by re-weighting the scores in SW-MSA. Could the authors provide more specific details on this operation? Is it a simple element-wise multiplication of the attention scores with the (processed) variance map, or is a more complex fusion mechanism employed?

---

> ### Author Response · Authors · 2025-11-22
> **Official Comment by Authors**
>
> > **Q1**: The meaning of the posterior mean (the "mode") remains less explored.
>
> **A1:** We appreciate this insightful suggestion. The interpretability of the latent space is indeed crucial. To address this, we have added a dedicated analysis in **Appendix A.7** (visualized in **Fig. 14**) to explicitly decode the behavior of the posterior mean.
> **1. Methodology:**
> We conduct a controlled perturbation experiment: we fix two degradation factors (e.g., noise and JPEG) and systematically varys the third (e.g., blur $\sigma \in [0.2, 4.0]$). We then track the 64-D posterior mean vectors and identify the top-10 most sensitive dimensions (those with the highest variance) for each factor.
>
> **2.Key Findings:**
> *    **Structured Encoding of Severity:** As shown in **Fig. 14(a)**, the dimensions sensitive to a specific factor do not fluctuate randomly. Instead, they exhibit **smooth, monotonic trajectories** that correlate perfectly with the degradation strength. This proves the mean is not an opaque code but an ordered, continuous representation.
> * **Factor Disentanglement (Subspace Separation):** As shown in **Fig. 14(b)**, the dimensions most sensitive to blur tend to differ from those responsive to noise or JPEG. This pattern suggests that the encoder organizes different degradation factors into **partially distinct subspaces**.
>
> **3. Conclusion:**
> In summary, the posterior mean is not a black box; it is a structured, interpretable representation where specific subspaces explicitly encode the type and intensity of distinct degradation factors.
>
> > **Q2**:Generalization may be limited when real-world degradations are mismatched with or underrepresented in the synthetic training set. How does the model perform on entirely out-of-distribution degradation types not seen during training (e.g., motion blur, atmospheric haze, or specific sensor noise patterns)? Does the probabilistic nature of SAVL provide enhanced robustness to such domain shifts compared to deterministic methods?
>
> **A2:** We appreciate this question regarding generalization. To address your concern about out-of-distribution (OOD) robustness, we have included a new section, **Appendix A.8**, along with **Figure 15**, to explicitly analyze the model's behavior on degradation types not seen during training.
>
> * **Generalization to Unseen Degradation Types (Motion Blur & Haze):**
>   We synthesize degradations entirely absent from our training set, specifically **motion blur** (varying kernel lengths) and **atmospheric haze/fog** (varying concentrations).
>     * **Visual Evidence (Fig. 15):** We utilize t-SNE to visualize the degradation representations extracted by SAVL and deterministic baselines (CDFormer, KDSR, LightBSR) on these OOD samples.
>     * **Result:** As shown in **Fig. 15**, SAVL successfully clusters these unseen degradations according to their severity levels (e.g., forming distinct progressions for different motion blur lengths). In contrast, deterministic baselines tend to produce scattered or collapsed embeddings, failing to distinguish between different levels of OOD degradations.
> * **Robustness to Unseen Sensor Noise (SVSR):**
> As noted in Appendix A.8, the **SVSR dataset** used in our main experiments effectively serves as an OOD test for sensor noise, as our training set contains no data from these specific sensors. Despite never seeing these sensor patterns during training, SAVL forms distinct clusters (as detailed in Sec 4.2 and Fig. 6), demonstrating its ability to generalize to novel noise distributions.
> * **Probabilistic vs. Deterministic Robustness:**
> Your hypothesis is supported by our findings. The experimental results in **Fig. 15** suggest that the **probabilistic nature of SAVL provides enhanced robustness** to domain shifts compared to deterministic methods.
> Deterministic baselines learn rigid point-to-point mappings that often overfit to the specific degradation characteristics of the training set. When encountering Out-Of-Distribution (OOD) data (e.g., Motion Blur), these rigid mappings fail to generalize, leading to the scattered or collapsed feature distributions observed in Fig. 15. In contrast, SAVL models degradation as a Gaussian distribution constrained by KL regularization. This variational constraint enforces a smoother and more continuous latent manifold. Consequently, even for unseen degradation types, SAVL can map variations in severity to structured, ordered trajectories in the latent space rather than mapping them randomly.

---

> ### Author Response · Authors · 2025-11-22
> **Official Comment by Authors**
>
> > **Q3**: A brief discussion on training stability, convergence speed, and the overall training time/cost compared to simpler implicit representation learning methods would be beneficial.
>
> **A3:** Thank you for the suggestion. We have added a discussion on training dynamics in **Appendix A.9**, supported by loss curves in **Figure 16**.
>
> * **Stability and Convergence:**
>     Our training process is stable. To prevent posterior collapse, we employ a **KL annealing strategy**, gradually increasing the KL weight over the first 50k iterations. As visualized in the TensorBoard curves in **Figure 16**, the reconstruction loss steadily decreases and converges around **200k iterations**. The KL loss and reconstruction loss effectively counterbalance each other, maintaining stable dynamics throughout optimization.
> * **Training Time and Cost:**
>     Our total training time is approximately **48 hours** for the degradation representation learning phase and another **48 hours** for the SR network training (on 8x RTX 3090 GPUs).
> * **Comparison with Implicit Methods:**
>     * ***vs.* Multi-stage Methods (*e.g.*, CDFormer, KDSR):** Since our framework follows a **two-stage paradigm** as CDFormer and KDSR, the training complexity and computational overhead are **at a comparable magnitude**. These methods similarly require separate optimization phases (e.g., teacher-student distillation or diffusion priors), placing them in the same computational tier as SAVL.
>     * ***vs.* Lightweight Methods (*e.g.*, DASR, LightBSR):** We acknowledge that our training cost is higher than single-stage approaches like DASR or lightweight approaches like LightBSR. However, this increased computational investment is a necessary trade-off to achieve the fine-grained spatial modeling and content decoupling that simpler methods lack, as demonstrated by our superior performance in complex real-world scenarios.
> > **Q4**: Did the authors experiment with using other direct MI estimators, such as CLUB, as a regularizer in the main objective ? How would integrating such an estimator directly into the SAVL framework compare to the current formulation?
>
> **A4:** Thank you for the question. Yes, we conduct experiment with using CLUB as a direct MI regularizer in the main objective.
>
> Specifically, we try to plug CLUB into SAVL to directly constrain MI terms such as $I(r; z)$ and $I(r; y)$. In practice, this formulation is difficult to optimize in our high-dimensional setting: the additional conditional predictor required by CLUB do not converge stably, which makes the overall training unstable and lead to consistently worse performance than our current SAVL objective. As a result, we decide not to use CLUB as a main regularizer, and instead keep it only as an ablation baseline (the “+CLUB” variants in Tables 1 and 2) to show that our stochastic SAVL formulation is more effective than deterministic representations even when they are equipped with a strong MI estimator like CLUB.
>
>
> > **Q5**: More specific details on e-weighting the scores in SW-MSA.
>
> **A5:** Thanks for your comments. Our SW-MSA indeed re-weights the attention map via simple element-wise multiplication. Specifically, we first derive a **severity map** from the variance map and use it to obtain additional query/key projections $(Q_s, K_s)$, which produce a **severity similarity** score map $A_s$. This severity attention map $A_s$ is then multiplied point-wise with the original SW-MSA attention scores $A_f$ to obtain the final attention map, followed by the standard normalization step.
> The whole process is a direct element-wise fusion, as illustrated on the right side of Fig. 3, rather than a more complex fusion mechanism. This design keeps the fusion mechanism simple and transparent, while allowing the variance/severity information to directly re-weight the spatial attention.

---

### Official Review · Reviewer_44sz · 2025-11-04

**Soundness:** 3
**Presentation:** 2
**Contribution:** 3
**Rating:** 6
**Confidence:** 3

**Summary:**

This paper aims to tackle two crucial issues in Single-Image Super-Resolution: inability to model spatially variant degradation and difficulty of degradation-content decoupling. To this end, it proposes Spatially Amortized Variational Learning (SAVL) algorithm, which models per-pixel degradations as spatially varying Gaussians inferred from local neighborhoods. Meanwhile, SAVL imposes an explicit mutual information constraint to suppress the degradation-content entanglement.

**Strengths:**

1. The paper is motivated well and modeling of spatially variant degradation is indeed crucial for single image super resolution.
2. The proposed Spatially Amortized Variational Learning (SAVL)  framework models per-pixel degradations as spatially varying Gaussians, which is a novel design.

**Weaknesses:**

1. More visual comparisons for ablation study are favorable to show the effectiveness of the proposed SAVL in 1) modeling the spatially variant degradations and 2) mitigating the degradation-content entanglement.

2. How to perform quantitative evaluation on the effectiveness of SAVL on degradation-content decoupling? It is indeed not straightforward to conduct such evaluation whilst it is important.

Minor issues:
Figure 1 and Figure 5 are too small to deliver a good presentation.

**Questions:**

Check the weaknesses.

---

> ### Author Response · Authors · 2025-11-22
> **Official Comment by Authors**
>
> > **Q1**: More visual comparisons for ablation study are favorable to show the effectiveness of the proposed SAVL in 1) modeling the spatially variant degradations and 2) mitigating the degradation-content entanglement.
>
> **A1:** We appreciate this constructive suggestion. **We have added comprehensive visual comparisons and quantitative ablation studies in the revised Appendix A.6 to explicitly demonstrate SAVL’s effectiveness.**
>
> **1. Effectiveness in Modeling Spatially Variant Degradations:**
> We perform t-SNE visualization and k-means clustering on pixel-level degradation features under spatially varying degradation settings in Appendix A.6.1.
>
> * **Cluster Separation (Fig. 11 & Table 5):** As shown in **Fig. 11**, our full SAVL model forms distinct, consistent clusters corresponding to specific local degradation types/levels. In contrast, deterministic representations (w/o SAVL) or those using alternative constraints (e.g., CLUB) fail to separate these local degradations, resulting in collapsed clusters. This is quantitatively confirmed by **Table 5**, where SAVL achieves significantly higher clustering accuracy (e.g., **72.44%** for variant types vs. 48.02% for w/o SAVL).
> * **Sensitivity vs. Brightness (Fig. 12):** Furthermore, **Fig. 12** reveals a critical insight: without SAVL, the learned deterministic features primarily follow the image brightness/texture (RGB values) rather than the actual degradation. SAVL effectively suppresses this correlation, capturing the true spatial variance of degradation severity.
>
> **2. Effectiveness in Mitigating Degradation-Content Entanglement:**
> Directly visualizing disentanglement is challenging, so we design a proxy hybrid reconstruction experiment in Appendix A.6.2. We feed the content encoder with a clean image $A$ and the degradation encoder with a degraded image $B$, then decode the result.
>
> * **Visual Evidence (Fig. 13):** If purely disentangled, the output should be "Content A + Degradation B".
>     * **w/o SAVL:** The output is heavily contaminated by the semantic content of image $B$, proving severe entanglement.
>     * **w/o SAVL + CLUB:** While slightly better, it still introduces noticeable edges and textures from image $B$.
>     * **Full SAVL:** Only our method produces a clean hybrid image that preserves the content of $A$ while faithfully rendering the degradation style of $B$, demonstrating superior content-degradation decoupling.

---

> ### Author Response · Authors · 2025-11-22
> **Official Comment by Authors**
>
> > **Q2**: How to perform quantitative evaluation on the effectiveness of SAVL on degradation-content decoupling? It is indeed not straightforward to conduct such evaluation whilst it is important.
>
> **A2:** We appreciate you highlighting this critical challenge. As you noted, evaluating decoupling is non-trivial. To address this, we formulate the evaluation as a statistical dependence analysis problem: a valid degradation representation should be **highly dependent on degradation factors** but **independent of content (scene) semantics**.
> We proceed in two steps and report the results in Table 1:
> **1. Linear Probing (Predictability Analysis):**
> We train lightweight linear classifiers on the frozen representations to predict ground-truth labels[1].
> * **Metric:** High accuracy for degradation labels (e.g., Noise Level) and low accuracy for content labels (e.g., Scene ID) indicate successful decoupling.
> * **Result:** As shown in **Table 1**, our method achieves a Noise-Level Accuracy of 94.80%, surpassing LightBSR (91.10%) and CDFormer (83.93%). Crucially, for content, our Scene-ID Accuracy drops to 49.97%, whereas baselines like KDSR and CDFormer retain high content predictability (>84%), indicating severe entanglement.
>
> **2. Dependence Measures (HSIC[2] & MINE[3]):**
>
> * **Maximal Degradation Dependence (Higher is Better):**
>     Our representation achieves the highest dependence scores with degradation factors on both metrics.
>     * **HSIC:** Ours (**0.0304**) outperforms the state-of-the-art baselines CDFormer (0.0270) and LightBSR (0.0133).
>     * **MINE:** Ours (**1.2920**) surpasses CDFormer (1.2694) and LightBSR (1.2425).
>     This confirms that SAVL captures the degradation signal among all the implicit methods.
>
> * **Minimal Content Dependence (Lower is Better):**
>     Our representation significantly suppresses content dependency compared to all the baselines.
>     * **MINE:** SAVL reduces the Mutual Information with Scene ID to **0.1507**, which is an order of magnitude lower than CDFormer (1.2209) and LightBSR (0.8823).
>     * **HSIC:** Crucially, the ablation study shows that removing SAVL nearly doubles the content dependency score (HSIC-SceneID increases from **0.0124** to **0.0220**), proving that our variational framework is the key driver for decoupling.
>
> **Conclusion:**
> These metrics quantitatively confirm that SAVL successfully maximizes degradation information retention while actively compressing irrelevant content information, validating the effectiveness of our proposed mutual-information suppression strategy.
>
> [1] Understanding intermediate layers using linear classifier probes.(ICLR2017)
> [2] A kernel statistical test of independence. (NIPS 2007)
> [3] MINE: Mutual Information Neural Estimation.(ICML2018)
>
> >**Minor issues**: Figure 1 and Figure 5 are too small to deliver a good presentation.
>
> **A:** Thank you for pointing this out. As suggested, we have improved them in the revised manuscript.

---

### Meta-Review · Area_Chair_73rS · 2026-01-05

**Summary:**

This paper tackles two crucial super-resolution issues: inability to model spatially variant degradation and difficulty of degradation-content decoupling. This work has five reviewers with three positive reviewers (8, 6, 6) and two negative reviewers (4, 4). In the rebuttal, the authors have provided many experiments to demonstrat that the proposed pixe-wise Gaussian representation can learn complex degradation patterns. Moreover, the authors compare with more SOTA and diffusion models., and clarify more issues. Although the two negative reviewers do not raise the scores, I agree with three positive reviewers to accept this work.

**Reviewer Concerns:**

Most of concerns have been addressed by the rebuttal. The authors are suggested to prepare the final version based on the rebuttal.

**Reviewer Scores:**

This work has five reviewers with three positive reviewers (8, 6, 6) and two negative reviewers (4, 4). All reviewers do not change the scores during the rebuttal.

---

### Decision · Program_Chairs · 2026-01-26

Accept (Poster)